# PROGRAMMATIC REINFORCEMENT LEARNING WITHOUT ORACLES

**Wenjie Qiu**
Department of Computer Science
Rutgers University
`wq37@cs.rutgers.edu`

**He Zhu**
Department of Computer Science
Rutgers University
`hz375@cs.rutgers.edu`

## ABSTRACT

Deep reinforcement learning (RL) has led to encouraging successes in many challenging control tasks. However, a deep RL model lacks interpretability due to the difficulty of identifying how the model's control logic relates to its network structure. Programmatic policies structured in more interpretable representations emerge as a promising solution. Yet two shortcomings remain: First, synthesizing programmatic policies requires optimizing over the discrete and non-differentiable search space of program architectures. Previous works are suboptimal because they only enumerate program architectures greedily guided by a pretrained RL oracle. Second, these works do not exploit compositionality, an important programming concept, to reuse and compose primitive functions to form a complex function for new tasks. Our first contribution is a programmatically interpretable RL framework that conducts program architecture search on top of a continuous relaxation of the architecture space defined by programming language grammar rules. Our algorithm allows policy architectures to be learned with policy parameters via bilevel optimization using efficient policy-gradient methods, and thus does not require a pretrained oracle. Our second contribution is improving programmatic policies to support compositionality by integrating primitive functions learned to grasp task-agnostic skills as a composite program to solve novel RL problems. Experiment results demonstrate that our algorithm excels in discovering optimal programmatic policies that are highly interpretable.

## 1 INTRODUCTION

A growing body of research has explored *programs* in a domain-specific programming language as a new RL policy representation that intentionally encourages policy interpretability. Yet, learning a policy as a high-level program in a structured representation is challenging. This is because algorithms must jointly identify a reasonable program architecture to allow for sufficient expressiveness while optimizing the parameters of the program modules in the architecture. For example, depending on the shape of a maze, walking a robot to different goals on the maze by a program may require various *if-then-else* conditions to travel along different paths, the number of which might not be known to the agent before training. To curb the non-differentiable program architecture search space, existing programmatic policy learning algorithms either learn from a *pretrained* program embedding space that must support smooth interpolation (Trivedi et al., 2021), or must be guided by the supervision of a *pretrained* oracle (e.g. a neural network policy trained by RL) via a teacher-student learning paradigm (Bastani et al., 2018; Silver et al., 2020; Inala et al., 2020; Verma et al., 2018; 2019). The task of imitating an oracle is significantly simpler than the full RL problem. However, since the policy space of neural networks and that of programmatic policies are very different, a significant performance gap exists between an imitating program and its RL oracle (Verma et al., 2018).

The first contribution of our paper is a framework to synthesize interpretable and *differentiable* programmatic policies solely from reward signals by policy gradient methods, without needing any oracles or *pretraining*. A conceivable way of synthesizing program architectures would be to enumerate all possible architectures induced by the grammar of a domain-specific policy language, run standard RL for each to find the optimal values of its unknown parameters, and return the best program. However, doing so is computationally expensive as each RL trial may explore millions

of environment steps. Inspired by recent advances in differentiable neural architecture search e.g. DARTs (Liu et al., 2019b), we relax the discrete program architecture search space to be continuous. Specifically, we encode program architecture synthesis as learning the probability distribution over all possible architecture derivations (up to a certain bound on program abstract syntax tree depth) induced by a policy language grammar. This enables our RL algorithm to jointly optimize program architectures and the parameters of program modules via policy-gradient methods.

Our second contribution is improving programmatic policies to support compositionality — the integration of primitive functions trained to grasp basic, task-agnostic skills (e.g. running forward or jumping) into a new complex function as a composite model to solve novel RL problems (e.g. jumping over multiple hurdles to reach a target). As opposed to policy ensemble models based on neural networks (Qureshi et al., 2020), our programmatically composite models interpret how primitive functions are composed under different environment conditions based on an RL agent's perceptions, and naturally generalize to novel scenarios. We further apply programmatic policies to address challenging hierarchical RL problems. Our solution leverages the specifications of primitive functions to create an optimal high-level control plan via a satisfiability constraint solver and implements the high-level plans by learned composition of primitive functions.

Finally, we benchmark our method against the state-of-the-art RL methods. Our results demonstrate that the programmatic RL framework is able to solve extremely hard RL problems using highly interpretable policies with improved task performance.

## 2 PROBLEM MOTIVATION AND FORMULATION

We study how to express RL policies as *differentiable* programs, which use symbolic language constructs to compose a set of parameterized primitive modules. To control an agent, a programmatic policy takes an environment state as input and computes an action as return for the agent to execute.

We view a programmatic policy as a pair $(E, \theta)$, where $E$ is a discrete program architecture and $\theta$ is a vector of real-valued parameters of the program. A program architecture $E$ is structured based on the context-free grammar (Hopcroft et al., 2007) of a policy DSL. In this paper, we consider the context-free grammar depicted in the standard Backus-Naur form (Winskel, 1993) in Fig. 1. A vertical bar "|" indicates choice. Such a grammar consists of a set of production rules $X ::= \sigma_1 \sigma_2 \cdots \sigma_j$ where $X$ is a nonterminal and $\sigma_1, \cdots, \sigma_j$ are either terminals or nonter-

$$E ::= C \mid \textbf{if } B \textbf{ then } C \textbf{ else } E$$
$$B ::= \theta_c + \theta^T \cdot \mathcal{X} \geq 0$$

Figure 1: A Context-free DSL Grammar for programmatic policies.

minals. For example, we may expand the nonterminal $E_1$ in a partial program **if** $B_1$ **then** $C_1$ **else** $E_1$ to **if** $B_1$ **then** $C_1$ **else** (**if** $B_2$ **then** $C_2$ **else** $E_2$). The nonterminals $E$ and $B$ stand for program expressions that evaluate to action values in $\mathbb{R}^m$ and Booleans, respectively, where $m$ is the action dimension size. We represent a state input to a programmatic policy as $s = \{x_1 : \nu_1, x_2 : \nu_2, \ldots, x_n : \nu_n\}$ where $n$ is the state dimension size and $\nu_i = s[x_i]$ is the value of $x_i$ in $s$. As usual, the unbounded variables in $\mathcal{X} = [x_1, x_2, \ldots, x_n]$ are assumed to be input variables (state variables in our context). A terminal in this grammar is a symbol that can appear in a program's code, e.g. the *if* symbol and $x_i$.

The semantics of a program in $E$ is mostly standard and given by a function $[\![E]\!](s)$, defined for each DSL construct. For example, $[\![x_i]\!](s) = s[x_i]$ reads the value of a variable $x_i$ in a state input $s$. A policy may use an **if-then-else** branching construct. To avoid discontinuities for differentiability, we interpret its semantics in terms of a smooth approximation where $\sigma$ is the sigmoid function:

$$[\![\textbf{if } B \textbf{ then } C \textbf{ else } E]\!](s) = \sigma([\![B]\!](s)) \cdot [\![C]\!](s) + (1 - \sigma([\![B]\!](s))) \cdot [\![E]\!](s) \qquad (1)$$

Thus, any policy programmed in this grammar becomes a differentiable program. $C$ is a controller used by a programmatic policy. During execution, the policy can invoke a set of controllers under different environment conditions, according to the activation of $B$ conditions in the program. We consider three DSLs depending on how $C$ is structured for *affine*, *ensemble*, and *PID* policies.

**Affine Policies.** The DSL for affine policies allows $C$ to be expanded as an affine transformation:

$$C_{\textit{Affine}} ::= \theta_c + \theta \cdot \mathcal{X} \mid \theta_c$$

> **if** $\theta_{1_c} + \theta_1^T \cdot \mathcal{X} > 0$
>     **then** $(\ 95\% \cdot \pi_{\text{UP}}(s) + 5\% \cdot \pi_{\text{LEFT}}(s))\ \longleftarrow$ **Branch 1**
>     **else if** $\theta_{2_c} + \theta_2^T \cdot \mathcal{X} > 0$
>         **then** $(\ 95\% \cdot \pi_{\text{LEFT}}(s) + 5\% \cdot \pi_{\text{RIGHT}}(s))\ \longleftarrow$ **Branch 2**
>         **else** $(\ 13\% \cdot \pi_{\text{DOWN}}(s) + 87\% \cdot \pi_{\text{RIGHT}}(s))\ \longleftarrow$ **Branch 3**
>
> $\mathcal{X} = [\ x,\ y,\ \mathcal{G}_x,\ \mathcal{G}_y, arctan\frac{y}{x}, \|x,y\|_2\ ]$
> $\theta_1 = [\ -2.052,\ 0.049,\ 0.440,\ 0.181,\ 0.241,\ 1.443\ ],\ \theta_{1_c} = -0.202$
> $\theta_2 = [\ 1.333,\ 2.204, -2.2171,\ 2.132,\ 1.878,\ 0.331\ ],\ \theta_{2_c} = -0.416$

Figure 3: An Ant Cross Maze program $\mathcal{P}_{cross}$ with three branches. A program input $\mathcal{X}$ includes current Ant position $x$, $y$ along with the target location $\mathcal{G}_x$, $\mathcal{G}_y$ (sampled from one of the three goals in Fig. 2). $arctan\frac{y}{x}$ and $\|x,y\|_2$ are functions of $x$ and $y$. Each branch composes primitive functions: $\pi_{\text{UP}}$, $\pi_{\text{DOWN}}$, $\pi_{\text{LEFT}}$, and $\pi_{\text{RIGHT}}$. Composition weights are shown in percentage.

where $\theta \in \mathbb{R}^{m \cdot |\mathcal{X}|}, \theta_c \in \mathbb{R}^m$ are policy parameters. Particularly, $C_{Affine}$ can be as simple as some (learned) constants $\theta_c$. An example affine policy is given in Appendix K.1.

**Ensemble Policies.** The most important feature of our programmatic model is compositionality — composing and reusing task-agnostic primitives in new programs to solve novel problems. The DSL for ensemble policies includes pre-acquired primitives $\pi_1, \cdots, \pi_N$ as callable library functions:

$$C_\pi \ ::= \ \theta_1 \cdot \pi_1(s) + \theta_2 \cdot \pi_2(s) + \cdots + \theta_N \cdot \pi_N(s)$$

$C_\pi$ explicitly compose primitive functions (e.g. running forward or jumping) hierarchically into a complex program (e.g. jumping over multiple hurdles to reach a target) where $\theta_1, \cdots, \theta_N \in \mathbb{R}^1$ parameterize a primitive combination. The input space of a primitive function can be different from that of a program (formally defined below). The semantics of $C_\pi$ is defined as follows:

$$[\![C_\pi]\!](s) = \sum_{i=0}^{N} q_i \cdot \pi_i(s) \text{ where } q_i = \frac{\exp(\theta_i/T)}{\sum_{j=0}^{N} \exp(\theta_j/T)}$$

Here the composition weights $\{q_i\}_{i=0}^N$ for primitive ensemble are computed using gumbel-softmax, where $T$ is the temperature term (Jang et al., 2017).

**PID Policies.** Suppose we know a priori that PID control is suitable for stabilising of an RL system. We can express this knowledge using the DSL for PID functions that allows $C$ to be expanded as discretized, multivariable PID controllers (Zheng et al., 2002). We leave the details in Appendix K.2.

**Program Interpretability.** Our RL algorithm searches over a DSL to synthesize programmatic policies. Thanks to their structured and symbolic representation, the algorithm learns highly interpretable policies. For example, consider an Ant Cross Maze environment depicted in Fig. 2. The maze contains three possible goal positions and one would be randomly selected at each time. In this environment, the task for a quadruped MuJoCo Ant is to reach the selected location by navigating through the maze staring from an initial position on the bottom and without collision or crash. We consider the DSL for this task using ensemble policies $C_\pi$. It includes four basic primitive functions for moving the Ant up $\pi_{\text{UP}}$, down $\pi_{\text{DOWN}}$, left $\pi_{\text{LEFT}}$, and right $\pi_{\text{RIGHT}}$ (pretrained as neural network policies using standard RL algorithms with details left in App. F.2).

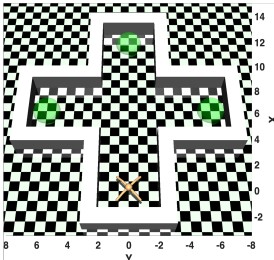

Figure 2: Ant Cross Maze

Fig. 3 depicts a synthesized program $\mathcal{P}_{cross}$ with three branches for solving the Ant Cross Maze environment. As specified in Equation 1, our semantics of a branching construct is approximated by the sigmoid function $\sigma$. The value of the predicate in a Boolean condition determines the activation of the controller guarded by the Boolean condition. At each state, branch activation determines the strength of each of the controllers in the program. For example, the activation of branch 1 is $\sigma(\theta_{1_c} + \theta_1^T \cdot \mathcal{X})$, and the activation of branch 2 is $(1 - \sigma(\theta_{1_c} + \theta_1^T \cdot \mathcal{X})) \cdot \sigma(\theta_{2_c} + \theta_2^T \cdot \mathcal{X})$.

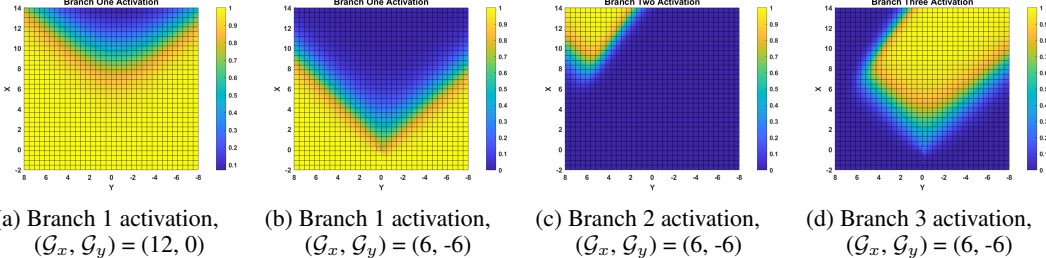

(a) Branch 1 activation,
$(\mathcal{G}_x, \mathcal{G}_y) = (12, 0)$

(b) Branch 1 activation,
$(\mathcal{G}_x, \mathcal{G}_y) = (6, -6)$

(c) Branch 2 activation,
$(\mathcal{G}_x, \mathcal{G}_y) = (6, -6)$

(d) Branch 3 activation,
$(\mathcal{G}_x, \mathcal{G}_y) = (6, -6)$

Figure 4: Branch activation as functions of Ant position $(x, y)$ for program $\mathcal{P}_{cross}$.

Fig. 4a depicts the activation of branch 1 as a function of $(x, y)$ when the goal to reach is sampled at $\mathcal{G}_x = 12, \mathcal{G}_y = 0$. The degree of activation (yellow) is close to 1 on all states under (12, 0) indicating that the ensemble policy at branch 1 is used to drive the Ant up to the goal. Indeed, according to the distribution of each primitive function at branch 1, the effect of $\pi_{\mathrm{UP}}$ dominates. Fig. 4b, Fig. 4c, and Fig. 4d depict the activation of all three branches when the goal is at $\mathcal{G}_x = 6, \mathcal{G}_y = -6$. The program can be interpreted as branch 1 (where $\pi_{\mathrm{UP}}$ dominates) and branch 3 (where $\pi_{\mathrm{RIGHT}}$ dominates) are activated in the yellow areas of Fig. 4b and Fig. 4d respectively. This allows the Ant to make a curved up and right move to the goal (branch 2 is not activated during execution for this goal).

**Problem Formulation.** We frame programmatic RL as a Markov Decision Process (MDP) defined by a tuple $\{\mathcal{S}, \mathcal{A}, \mathcal{T}, \mathcal{R}\}$ where $\mathcal{S}$ and $\mathcal{A}$ represent the environment state space and action space, $\mathcal{T} : \mathcal{S} \times \mathcal{A} \times \mathcal{S} \to [0, 1]$ captures the set of transition probabilities, and $\mathcal{R} : \mathcal{S} \times \mathcal{A} \to \mathbb{R}$ denotes the reward function. We assume $\mathcal{S} \supseteq \mathbb{R}^{|\mathcal{X} \cup \mathcal{V}|}$ where $\mathcal{X}$ is the set of input variables of a composite program (defined by a DSL) and $\mathcal{V}$ is the set of input variables of primitive functions. For an affine policy, $\mathcal{V} = \emptyset$. At time $t \geq 0$, an RL agent receives an environment state $s_t \in \mathcal{S}$ and performs an action $a_t \in \mathcal{A}$ selected by its policy $\pi(a_t|s_t) : \mathcal{S} \to \mathcal{A}$. Based on $s_t$ and $a_t$, the agent transits to receive the next state according to the transition model $\mathcal{T}(s_{t+1}|s_t, a_t)$, and receives the reward $R(s_t, a_t)$. We aim to learn a programmatic policy $\pi$ in the DSL in Fig. 1 by jointly synthesizing the program's architecture $E$ and optimizing the program's parameters $\theta$ to maximize the cumulative discounted reward $\mathbb{E}_{s_0, a_0, s_1 \cdots \sim \pi} \left[ \sum_0^\infty \gamma^t \cdot R(s_t, a_t) \right]$ where $\gamma \in (0, 1]$.

## 3 ARCHITECTURE SEARCH FOR PROGRAMMATIC POLICIES

Inspired by differentiable neural architecture search e.g. DARTs (Liu et al., 2019b), we relax the policy architecture search space to be continuous. This amounts to collectively optimizing the probability distribution of all program architectures in the search space and assigning the the highest probability to the architecture that maximizes cumulative MDP reward.

Our algorithm is not specific to a DSL. It takes as input any policy DSL with differentiable semantics and conducts policy architecture search on a *program derivation tree* of the DSL. Formally, a program derivation tree is $\mathcal{T} = \{V, \mathcal{E}\}$ where a node $u \in V$ contains partial architectures with missing expressions or a complete architecture permissible by the DSL. An edge $(u, u_E) \in \mathcal{E}$ exists if one can obtain the architectures in $u_E$ by expanding a nonterminal $E$ within a partial architecture in $u$ following some DSL production rules. If more than one rule can be applied to expand the nonterminal $E$, $u_E$ contains more than one architecture. Take the $\mathcal{P}_{cross}$ program in Fig. 3 as a concrete example: Fig. 5 depicts a program derivation tree for the DSL in Fig. 1 where a controller $C$ is an ensemble policy. On the root node 0, we have two choices to expand the initial nonterminal $E_1$ to either an ensemble policy $C_1$ or a partial architecture if $B_1$ then $C_2$ else $E_2$. Node 1 thus contains two partial architectures. Formally, we use $\mathcal{F}(u_E)$ to represent the set of architectures on a node $u_E$.

To expand a nonterminal or a missing expression of a partial program architecture, we relax the categorical choice of DSL production rules into a softmax over all possible production rules for the missing expression with trainable weights. For example, on node 1, the choices to expand $E_1$ between the ensemble policy $C_1$ and the conditional branching expression are weighted by the weight matrix $w_1$ (obtained after softmax) drawn in Fig. 5. Based on $w_1$, we choose to expand $E_1$ to the conditional branching expression on node 1. Assume we further expand $E_2$ on node 1 to a conditional branching expression as well on node 5. Then again we have two choices to expand the nonterminal $E_3$ on node 5 weighted by $w_2$. This time we choose to expand $E_3$ to an ensemble policy $C_5$. Formally, the

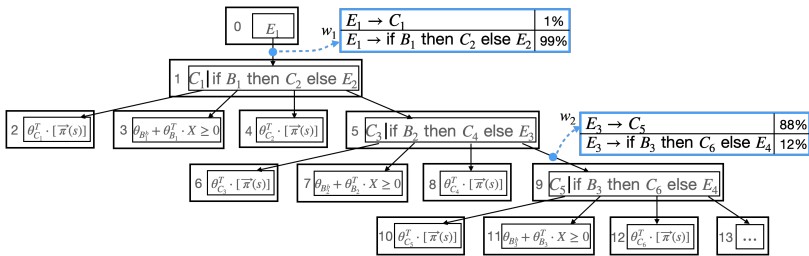

Figure 5: Ant Cross Maze Program Derivation Tree with program input $\mathcal{X}$. $\vec{\pi}$ refers to the primitives of Ant moving up $\pi_{\text{UP}}$, down $\pi_{\text{DOWN}}$, left $\pi_{\text{LEFT}}$ and right $\pi_{\text{RIGHT}}$ that take the Ant's own observations.

weight matrix $w_{u_E}$ of the incoming edge to a node $u_E$ is of the shape $\mathbb{R}^{|\mathcal{F}(u_E)|}$, and $w_{u_E}[E']$ weighs the likelihood of choosing a particular architecture $E' \in \mathcal{F}(u_E)$ for expanding $E$.

A program derivation tree $\mathcal{T}$ essentially expresses all possible program derivations up to a certain bound on the depth of program abstract syntax trees. To train architecture weights, we encode a program derivation tree itself as a differentiable program $\pi_{\theta,w}^{\mathcal{T}}$ that takes a state $s$ as input. Its action output is weighted by the outputs of all programs included in $\pi_{\theta,w}^{\mathcal{T}}$, where $w$ represents program architecture weights and $\theta$ includes unknown program parameters of all the mixed programs in the tree. The semantics computation of an expression $[\![E]\!](s)$ in a program derivation tree $\pi_{\theta,w}^{\mathcal{T}}$ is delegated to its tree node $u_E$ where the nonterminal $E$ is expanded and the categorical choice of expanding $E$ on $u_E$ is relaxed to a softmax over all possible choices:

$$[\![E]\!](s) = [\![u_E]\!](s) \qquad [\![u_E]\!](s) = \sum_{E' \in \mathcal{F}(u_E)} \frac{\exp(w_{u_E}[E'])}{\sum_{E'' \in F(u_E)} \exp(w_{u_E}[E''])} \cdot [\![E']\!](s)$$

**Complexity.** Assume that the root of $\mathcal{T}$ hosts the initial DSL nonterminal $E_{\mathcal{T}}$, $d$ is the depth of $\mathcal{T}$, $k$ is the number of DSL production rules, and $m$ is the maximum number of nonterminals in the body of any rules. The semantics of $\pi_{\theta,w}^{\mathcal{T}}$ is defined as $[\![\pi_{\theta,w}^{\mathcal{T}}]\!](s) = [\![E_{\mathcal{T}}]\!](s)$. The number of DSL operations (e.g. evaluations of ensemble policies and Boolean conditions) invoked by $[\![E_{\mathcal{T}}]\!](\cdot)$ is bounded by $O((km)^d)$. In practice, we optimize the run-time cost of $[\![E_{\mathcal{T}}]\!](\cdot)$ as discussed in Appendix. F.1.

The parameters $w$ and $\theta$ of a program derivation tree $\pi_{\theta,w}^{\mathcal{T}}$ can be jointly optimized using any policy gradient methods. To obtain stochastic policy gradients, $\pi_{\theta,w}^{\mathcal{T}}(\cdot|s)$ is encoded as a Gaussian policy where the tree program outputs the action distribution mean. A separate set of parameters specify the (diagonal) distribution covariance. In this paper, we consider trust region methods e.g. (Schulman et al., 2015) and aim to maximize the "surrogate" objective function, subject to a constraint on the size of the policy update by $\delta$, where $\rho_{\pi_{\theta_{old},w_{old}}^{\mathcal{T}}}$ is the discounted state visitation frequency of $\pi_{\theta_{old},w_{old}}^{\mathcal{T}}$, $A_{\pi_{\theta_{old},w_{old}}^{\mathcal{T}}}$ is an estimator of the advantage function over a finite batch of samples from $\pi_{\theta_{old},w_{old}}^{\mathcal{T}}$ and $\theta_{old}$, $w_{old}$ are policy parameters and architecture weights before the update:

$$\text{maximize}_{\theta,w} \; J_{\theta_{old},w_{old}}(\theta, w) = \mathbb{E}_{s \sim \rho_{\pi_{\theta_{old},w_{old}}^{\mathcal{T}}}, a \sim \pi_{\theta_{old},w_{old}}^{\mathcal{T}}} \left[ \frac{\pi_{\theta,w}^{\mathcal{T}}(s,a)}{\pi_{\theta_{old},w_{old}}^{\mathcal{T}}(s,a)} A_{\pi_{\theta_{old},w_{old}}^{\mathcal{T}}}(s,a) \right]$$

$$\text{subject to } \mathbb{E}_{s \sim \rho_{\pi_{\theta_{old},w_{old}}^{\mathcal{T}}}} \left[ D_{KL}(\pi_{\theta_{old},w_{old}}^{\mathcal{T}}(\cdot|s) \,\big|\big|\, \pi_{\theta,w}^{\mathcal{T}}(\cdot|s)) \right] \leq \delta \tag{2}$$

**Policy Parameter Optimization.** Our training algorithm is an iterative bilevel optimization procedure. At training iteration $k$, we perform two steps. At the first step, we optimize the lower-level program parameters $\theta$ with respect to (2), freezing the upper-level architecture weights $w$:

$$\theta_{k+1} = \arg\max_{\theta} J_{\theta_k,w_k}(\theta, w_k) \text{ s.t. } \mathbb{E}_{s \sim \rho_{\pi_{\theta_k,w_k}^{\mathcal{T}}}} \left[ D_{KL}(\pi_{\theta_k,w_k}^{\mathcal{T}}(\cdot|s) \,\big|\big|\, \pi_{\theta_{k+1},w_k}^{\mathcal{T}}(\cdot|s)) \right] \leq \delta \tag{3}$$

**Policy Architecture Optimization.** At the second step, we optimize the upper-level architecture weights $w$ with respect to (2), freezing the lower-level program parameters $\theta$:

$$w_{k+1} = \arg\max_{w} J_{\theta_{k+1},w_k}(\theta_{k+1}, w) \text{ s.t. } \mathbb{E}_{s \sim \rho_{\pi_{\theta_{k+1},w_k}^{\mathcal{T}}}} \left[ D_{KL}(\pi_{\theta_{k+1},w_k}^{\mathcal{T}}(\cdot|s) \,\big|\big|\, \pi_{\theta_{k+1},w_{k+1}}^{\mathcal{T}}(\cdot|s)) \right] \leq \delta$$
$$\tag{4}$$

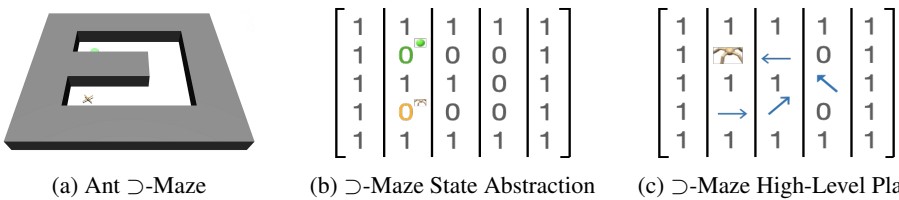

(a) Ant ⊃-Maze      (b) ⊃-Maze State Abstraction      (c) ⊃-Maze High-Level Plan

Figure 6: Model-based high-level planning for Ant ⊃-Maze.

Training steps (3) and (4) are alternated across training iterations until reward convergence. They can be approximately solved using the efficient conjugate gradient algorithm, after making a linear approximation to the objective and a quadratic approximation to the constraint (Schulman et al., 2015). Upon convergence, based on architecture weights, we obtain a discrete program architecture from $\pi_{\theta,w}^{\mathcal{T}}$ replacing each tree node containing multiple architectures with the most likely architecture in a top-down manner. Finally, we train the parameters in the chosen architecture using RL (Schulman et al., 2015) until convergence from the parameter values learned by the architecture search process. The algorithmic pseudocode is depicted in Algorithm 1 in the appendix.

## 4   PROGRAMMATIC HIGH-LEVEL PLANNING

We further explore learning programmatic policies for *high-level planning* tasks with long-horizon and weak reward signals. For example, consider the Ant navigation task in a ⊃-maze from (Nachum et al., 2018) in Fig. 6a. Due to the complex maze shape, distance-based rewards do not lead to solving the problem. Hierarchical RL (HRL) is an efficient approach to solve long-horizon high-level planning. HRL uses a high-level policy to generate a sequence of high-level goals and low-level policies to generate sequences of actions to achieve each successive goal. Existing HRL methods are either model-free e.g. (Nachum et al., 2018; 2019) or model-based e.g. (Roderick et al., 2018; Jothimurugan et al., 2021). In a model-free approach, the learning algorithm in Sec. 3 can be applied to train HRL policies as programs. In this section, we focus on model-based planning that takes advantage of the structure in the model of a high-level planning task to improve learning efficiency.

For model-based planning, we abstract over both states and actions to ensure a finite high-level model. Recall that functions in the DSL of ensemble policies are assumed to be simple, low-level primitives designed to grasp basic, task-agnostic skills (e.g. Ant moving up and down). We use these primitives as abstract actions because they abstract low-level agent actions to an abstract space of skills. Regarding state abstractions, similar to previous works (Abel et al., 2020; Winder et al., 2020; Jothimurugan et al., 2021), we assume that state abstractions can be effectively provided by domain experts. For instance, the abstract states of the ⊃-maze are collected on an abstract 2-D gridworld $m \in \mathbb{R}^{N \times N}$ in Fig. 6b. An abstract grid $(x, y) \in N \times N$ subsumes concrete positions with a known scale where $m[x, y] = 1$ indicates walls and $m[x, y] = 0$ indicates navigable spaces. The domain expert can then specify a pair of abstract initial state $\varphi_{\text{Init}}$ (yellow) and goal state $\varphi_{\text{Goal}}$ (green).

$$\varphi_{\text{Init}}(x_0, y_0) \equiv x_0 = 3 \wedge y_0 = 1 \qquad \varphi_{\text{Goal}}(x_k, y_k) \equiv x_k = 1 \wedge y_k = 1$$

Our design choice of state and action abstraction addresses an important challenge of model-based HRL — it has to simultaneously perform model-based high-level planning and estimate the state transition probabilities of the abstract model based on the current high-level policy (Jothimurugan et al., 2021). In our approach, we decouple this dependency. Since abstract actions as primitive functions implement simple, task-agnostic skills, it is straightforward to directly specify the state-transition behavior of the abstract actions $\varphi_{\text{Act}}$ as opposed to learning it. For example, the Ant primitives move an Ant by updating its horizontal position $x_i$ or vertical position $y_i$ one step $dx_i$ or $dy_i$ on the abstract model. The Ant can also be moved diagonally via primitive composition:

$$\varphi_{\text{Act}}(x_{i-1}, y_{i-1}, x_i, y_i) \equiv \exists dx_i, dy_i.\ x_i = x_{i-1} + dx_i \ \wedge \ y_i = y_{i-1} + dy_i \ \wedge -1 \le dx_i, dy_i \le 1$$

An abstract action is specified together with a guard $\varphi_{\text{Guard}}$ that encodes the condition under which the action can occur. For example, an Ant in ⊃-maze should not walk into walls and the path from an old position to a new position should not be blocked by walls.

$$\varphi_{\text{Guard}}(x_{i-1}, y_{i-1}, x_i, y_i) \equiv m[x_i][y_i] \ne 1 \ \wedge \big( m[x_{i-1}][y_i] \ne 1 \vee m[x_i][y_{i-1}] \ne 1 \big)$$

We further leverage the primitive function specifications to synthesize a high-level control plan on the abstract model. Suppose for now that we are searching over a high-level plan of a fixed number $k$ of abstract action steps to reach the goal state $\varphi_{\text{Goal}}$ from the initial state $\varphi_{\text{Init}}$:

$$\varphi_{\text{Init}}(x_0, y_0) \wedge \bigwedge_{i=1}^{k} \big( \varphi_{\text{Act}}(x_{i-1}, y_{i-1}, x_i, y_i) \wedge \varphi_{\text{Guard}}(x_{i-1}, y_{i-1}, x_i, y_i) \big) \wedge \varphi_{\text{Goal}}(x_k, y_k) \quad (5)$$

Constraint (5) can be solved by off-the-shelf constraint solvers e.g. Z3 (de Moura & Bjørner, 2008). We incrementally search for longer and longer high-level plans, starting from $k = 1$ and increasing $k$ until the optimal solution is found. Such a plan is a sequence of sub tasks. For example, the solved high-level plan for Ant ⊃-Maze navigation is depicted in Fig. 6c. Since a sub task may depend on multiple primitives e.g. Ant turning around at a corner, we learn a low-level policy for each sub task as a programmatic ensemble policy composing primitive functions using the technique in Sec. 3 guided by distance-based rewards. This ensures that our HRL policies are interpretable at both high level and low level. More details including the algorithmic pseudocode are given in Appendix. A.2.

Compared to existing model-based HRL approaches e.g. (Jothimurugan et al., 2021), our algorithm additionally needs abstract actions to be specified (as $\varphi_{\text{Act}}$ and $\varphi_{\text{Guard}}$) rather than learned. We show that (1) providing specifications of task-agnostic abstract actions (primitive functions) is trivial even for complex high-level planning tasks (Appendix G.1); (2) more importantly, these easy-to-annotate specifications substantially improve the sample efficiency of HRL (Sec. 5).

## 5 EXPERIMENTS AND EVALUATIONS

We evaluated our approach on two groups of challenging continuous control benchmarks involving motion control and task planning [1]. Group one contains four MuJoCo environments that require agents to reach or move an object to target locations: (1) **Ant Cross Maze**: the example depicted in Fig. 2. (2) **Ant Random Goal**: The quadruped MuJoCo Ant in Fig. 10a is trained to reach a randomly sampled goal location within a confined circular region. (3) **Pusher**: A robotic arm in Fig. 10b is trained to push a cylinder object to a given target location. (4) **HalfCheetah Hurdle**: A MuJoCo halfcheetah in Fig. 10c is required to run and jump over three hurdles to reach a given goal area. Group two consists of three hierarchical RL benchmarks: (1) **Ant ⊃-Maze**: the example depicted in Fig. 6a. (2) **Ant Push**: This task requires the Ant in Fig. 10d to push away a movable block to reach the goal region behind it. (3) **Ant Fall**: The Ant in Fig. 10e is required to push a movable block into a rift to fill the gap and then walk across it to reach the target on the other side of the rift. Multiple neural network primitives were trained and included in our ensemble policy DSL e.g. HalfCheetah $\pi_{\text{JUMP}}$ and $\pi_{\text{FORWARD}}$, Pusher $\pi_{\text{PUSH-LEFT}}$ and $\pi_{\text{PUSH-DOWN}}$ (with details in Appendix F.2).

**Programmatic RL.** To the benchmarks from the first group, we apply the RL algorithm in Sec. 3 to learn a program derivation tree $\pi^{\mathcal{T}}$-PRL from which we extract programmatic RL policies $\pi$-PRL. The tree depth bound of $\pi^{\mathcal{T}}$-PRL is set to 6. Our baselines include Composition-SAC (Qureshi et al., 2020) that learns Bidirectional LSTM-based ensemble policies using the SAC algorithm (Haarnoja et al., 2018) to combine the task-agnostic primitives for solving the task environments. On-policy RL algorithms TRPO (Schulman et al., 2015) and PPO (Schulman et al., 2017) are the other two baselines. Fig. 7a depicts the mean learning performance in terms of the agent's final distance from targets over 10 random seeds. TRPO and PPO both fail to reach the goals. During policy architecture search, $\pi^{\mathcal{T}}$-PRL successfully solves all tasks and outperform or match the performance of the BiLSTM models of Composition-SAC. The $\pi^{\mathcal{T}}$-PRL models exhibit less data-efficiency than the Composition-SAC policies in part due to the use of TRPO for on-policy update that is less sample-efficient than off-policy SAC (our learning algorithm can also be applied to off-policy RL).

Upon convergence of training a $\pi^{\mathcal{T}}$-PRL program derivation tree, our algorithm extracts a program $\pi$-PRL from $\pi^{\mathcal{T}}$-PRL and trains it until convergence. Table 1 presents the mean and standard deviation of $\pi$-PRL's final distance from given targets. Extracted programs recover the performance of $\pi^{\mathcal{T}}$-PRL models and continue to outperform or match the BiLSTM models in Composition-SAC. Importantly, $\pi$-PRL policies are significantly more interpretable than the BiLSTM policies e.g. Fig. 3 and Fig. 4. The high interpretability of $\pi$-PRL policies in turn leads to strong generalizability to novel environments because of the inductive bias implicitly encoded in the policy's interpretable

---

[1]Code is available at https://github.com/RU-Automated-Reasoning-Group/pi-PRL.

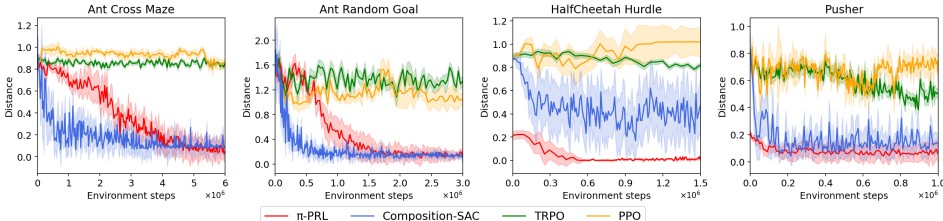

(a) Comparison with baselines for group one environments. Results are averaged over 10 random seeds.

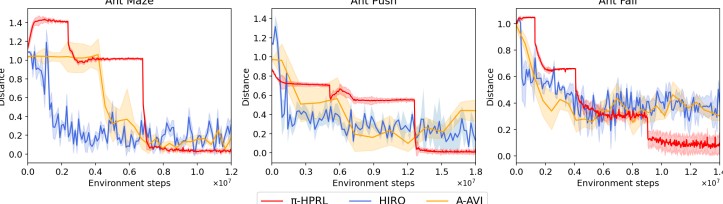

(b) Comparison with baselines for group two environments. Results are averaged over 5 random seeds.

Figure 7: Comparison results against baselines. The $y$-axis records the agent's distance towards its goal normalized by the agent initial distance from the goal so values close to 1 or higher show failures.

| Environment | Performance | | Convergence | | | Depth |
|---|---|---|---|---|---|---|
| | $\pi$-PRL | Comp-SAC | $\pi^{\mathcal{T}}$-PRL | $\pi$-PRL | Scratch | (max 6) |
| Ant Cross Maze | **0.10±0.06** | 0.11±0.05 | 5.0 | 3.0 | 4.0 | 4.75 |
| Ant Random Goal | 0.14±0.04 | **0.12±0.02** | 2.0 | 1.5 | 4.0 | 5.30 |
| HalfCheetah Hurdle | **0.03±0.01** | 0.43±0.22 | 0.5 | 0.0 | 1.0 | 3.10 |
| Pusher | **0.09±0.05** | 0.16±0.06 | 0.2 | 0.1 | 0.35 | 3.50 |

Table 1: Performance and convergence comparison for group one environments averaged over 10 random seeds. The performance section compares the mean normalized final distances to goals of extracted programs $\pi$-PRL with standard deviations; the convergence section compares averaged numbers of environment steps (in millions) until convergence for policy architecture search $\pi^{\mathcal{T}}$-PRL, policy extraction $\pi$-PRL, and training a program on the same architecture as $\pi$-PRL from scratch. The depth section shows mean abstract syntax tree depth of extracted $\pi$-PRL programs.

representation, e.g. if-then-else conditions used to drive the Ant to different regions on a maze. We show more examples about the interpretability and generalizability of $\pi$-PRL in Appendix I and J.

In Table 1, we also show the number environment steps needed until convergence for policy architecture search $\pi^{\mathcal{T}}$-PRL, extraction $\pi$-PRL, and training a programmatic policy using the same architecture as $\pi$-PRL from *scratch*. Except Ant Cross Maze, the sample efficiency of policy architecture search and extraction combined is better than learning a *single* program from scratch. Intuitively, this is because $\pi^{\mathcal{T}}$-PRL is essentially an over-parameterized model of the optimal policy in the search space. The better data efficiency of our method compared to enumerating all possible programs to perform RL for each highlights the merits of jointly optimizing policy architectures and parameters.

**Programmatic High-level Planning**. To the second group of the benchmarks, we apply the high-level planning algorithm in Sec. 4 to learn hierarchically programmatic RL policies, denoted as $\pi$-HPRL programs. Our baselines include a model-free Hierarchical RL method HIRO (Nachum et al., 2018) and a model-based high-level planning method A-AVI (Jothimurugan et al., 2021). All algorithms run under the same setting to reach a given target, which is fixed for each environment. Fig. 7b depicts the mean learning performance in terms of the agent's final distance from targets over 5 random seeds. The $\pi$-HPRL programs are more stable, outperform the baselines, and uniquely solve the Ant Push and Ant Fall environments. We show the success rates of each algorithm in Appendix G.2. Fig. 7b shows that HIRO gets stuck in local optimums. Unlike $\pi$-HPRL, HIRO is model-free and does not know the structure of the abstract state space, so it is unable to discover the path from the initial region to the goal region. A-AVI performs worse than $\pi$-HPRL because it does *not* decouple model-based high-level planning and model construction. $\pi$-HPRL is more sample efficient as it decomposes a task into multiple simpler sub tasks with explicit sub goals to

| Environment | $\pi$-Affine | $\pi^5$-Ensem | | $\pi^6$-Ensem | | $\pi^7$-Ensem | | $\pi$-Oracle |
|---|---|---|---|---|---|---|---|---|
| | Distance | Distance | $d$ | Distance | $d$ | Distance | $d$ | Distance |
| Ant Cross Maze | 0.87±0.03 | 0.10±0.06 | 4.2 | 0.10±0.06 | 4.8 | **0.07±0.05** | 5.6 | 0.33±0.16 |
| Ant Random Goal | 0.95±0.06 | 0.16±0.03 | 4.4 | **0.14±0.04** | 5.3 | 0.15±0.05 | 5.2 | 0.42±0.22 |
| HalfCheetah Hurdle | 0.57±0.13 | 0.03±0.01 | 3.1 | **0.03±0.01** | 3.1 | 0.04±0.02 | 2.6 | 0.67±0.24 |
| Pusher | 0.36±0.05 | **0.08±0.04** | 3.0 | 0.09±0.05 | 3.5 | 0.10±0.04 | 4.0 | 0.26±0.17 |

Table 2: Ablation study on different configurations of our algorithm. $\pi$-Affine denotes a non-compositional programmatic policy with *affine* controllers (the depth bound of its program derivation tree is set to 6). $\pi^{depth}$-Ensem represents a compositional program with primitive *ensembles* learned by setting the corresponding program derivation tree $depth$ bound. $\pi$-oracle denotes a policy learned by imitating its oracle (Verma et al., 2018). $d$ shows mean abstract syntax tree depth of learned policies. The mean normalized final distances to goals are averaged over 10 random seeds.

reach in an optimal high-level plan. First, since the high-level plan is in accordance with the primitive specifications, a sub goal is easy to achieve by utilizing the corresponding primitives. Second, the training procedure can be guided by dense distance-based rewards to the sub goal. Therefore, training for each sub task converges extremely fast well in advance of the expiration of the training budget allocated to the sub task, causing the ladder-shape convergence curves in Fig. 7b.

**Ablation Study**. We investigate the impact of compositionality in our algorithm. For the first group of our benchmarks, we apply our algorithm to synthesize programs in the DSL in Fig. 1 that switch back and forth between a set of *affine* controllers under different conditions. Table 2 shows that these programs perform much worse than the composite programs, which highlights the merits of composition. We additionally study the effect of the depth bound on a program derivation tree for policy architecture search. We set the depth bound to 5, 6, 7 respectively. Table 2 shows that the final performance of the synthesized policies is not sensitive to these settings. Our algorithm converges to similar architectures as it strives to assign the highest probability to the architecture that maximizes cumulative RL reward. Convergence curves of all the ablated versions are depicted in Appendix B.

Table 2 also reports the performance comparison between our oracle-free programmatic RL algorithm with an oracle-guided baseline (Verma et al., 2018). The baseline learns programmatic policies that imitate the neural Compositional-SAC policies in Table 1. To ensure a fair comparison, we relax the semantics of these programs to be continuous as well. The results show that the performance of the distilled programs is worse than that of their oracles and our policies. Since the neural oracles and programs reside in very different policy architecture spaces, the program that best imitates an oracle is not necessarily a performant programmatic policy. Comparison results with other oracle-based programmatic RL baselines can be found in Appendix C.

## 6 RELATED WORK AND CONCLUSION

Existing programmatic RL methods mostly train a programmatic policy to imitate a pretrained RL oracle. They synthesize policies by enumerating a set of templates in the form of decision trees (Bastani et al., 2018; Silver et al., 2020), finite state machines (Inala et al., 2020), or program sketches (Verma et al., 2018; 2019). However, imitation-based approaches suffer from a nontrivial distillation gap as the distillation process can yield suboptimal policies. Our approach optimizes programmatic polices in a continuous relaxation of the non-differentiable architecture space solely using reward signals by policy-gradient methods. Trivedi et al. (2021) first learns a smooth program embedding space and then searches over the embedding space to synthesize a program. Our approach differs as we do not need to prepare a dataset of programs to train the program embedding space, thus is more suitable when sampling such a dataset is challenging. Yang et al. (2021) uses a MaxSAT solver to synthesize straight-line programs to guide a policy to reach a goal. Our method is more suitable when there are multiple goals for which architecture synthesis is necessary to learn conditional branches to reach different goals. We discuss other related work in broader contexts in Appendix. E.

**Conclusion.** We present a novel programmatically interpretable and compositional RL framework. Our method jointly learns policy parameters and policy architectures in a continuous relaxation of the non-differentiable program architecture space via policy-gradient methods. Our RL framework leverages compositionality in programming languages to integrate primitive functions into a composite program to solve novel RL problems. Experiment results demonstrate that it excels in discovering compositional RL programs with optimal architectures and strong interpretability.

## REPRODUCIBILITY STATEMENT

We have included instructions to reproduce our results in the supplementary material. The code of this work is available at https://github.com/RU-Automated-Reasoning-Group/pi-PRL.

## ACKNOWLEDGMENTS

We thank the anonymous reviewers for their comments and suggestions. This work was supported by NSF Award #CCF-2124155 and NSF Award #CCF-2007799.

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

## A    TRAINING ALGORITHM PSEUDOCODE

### A.1    THE PROGRAMMATIC REINFORCEMENT LEARNING ALGORITHM

We depict the pseudocode of our oracle-free programmatic reinforcement learning algorithm in Algorithm 1. The algorithm builds a program derivation tree that involves programs whose abstract syntax tree depth is bounded by a hyperparameter $d$. We encode a program derivation tree as a differentiable program $\pi_{\theta,w}^{\mathcal{T}}$ with architecture weight $w$ and policy parameters $\theta$. Recall the surrogate objective definition in Eq. (2):

$$J_{\theta_{old},w_{old}}(\theta,w) = \mathbb{E}_{s\sim\rho_{\pi_{\theta_{old},w_{old}}^{\mathcal{T}}},a\sim\pi_{\theta_{old},w_{old}}^{\mathcal{T}}}\left[\frac{\pi_{\theta,w}^{\mathcal{T}}(s,a)}{\pi_{\theta_{old},w_{old}}^{\mathcal{T}}(s,a)}A_{\pi_{\theta_{old},w_{old}}^{\mathcal{T}}}(s,a)\right]$$

where $A_{\pi_{\theta_{old},w_{old}}^{\mathcal{T}}}$ is an estimator of the advantage function over a finite batch of samples from $\pi_{\theta_{old},w_{old}}^{\mathcal{T}}$ and $\theta_{old}$, $w_{old}$ are policy architecture weights and parameters before policy update.

In the pseudocode, training steps (3) for learning $\theta$ and (4) for learning $w$ are alternated across training iterations until reward convergence. They can be approximately solved using the efficient conjugate gradient algorithm, after making a linear approximation to the objective and a quadratic approximation to the constraint (Schulman et al., 2015). Upon convergence, based on architecture weights, we obtain a discrete program architecture from $\pi_{\theta,w}^{\mathcal{T}}$ replacing each tree node containing multiple architectures with the most likely architecture in a top-down manner. Finally, we train the parameters in the chosen architecture using RL (Schulman et al., 2015) until convergence from the parameter values learned by the search process.

---

**Algorithm 1** Programmatic Reinforcement Learning without Oracles

---

**Require:** Policy Context-free grammar $E$, depth bound $d$, KL-divergence limit $\delta$
**Ensure:** Synthesized programmatic policy $\pi$
    $\mathcal{T} \leftarrow$ get_Program_Derivation_Tree $(E,d)$
                    $\triangleright$ $\mathcal{T}$ expresses all programs whose abstract syntax tree depth is bounded by $d$
    $\pi_{\theta,w}^{\mathcal{T}} \leftarrow$ encode $(\mathcal{T})$
        $\triangleright$ $\pi_{\theta,w}^{\mathcal{T}}$ as differentiable program with parameters $\theta$ and architecture weights $w$ that encodes $\mathcal{T}$
    $\theta_0, w_0 \leftarrow$ randomly_initialize $(\theta,w)$
    **for** $k = 0,1,2,\ldots,M$ **do**                    $\triangleright$ Policy architecture search
        $\theta_{k+1} = \arg\max_{\theta} J_{\theta_k,w_k}(\theta,w_k)$
        s.t. $\mathbb{E}_{s\sim\rho_{\pi_{\theta_k,w_k}^{\mathcal{T}}}}\left[D_{KL}(\pi_{\theta_k,w_k}^{\mathcal{T}}(\cdot|s) \,\big|\big|\, \pi_{\theta_{k+1},w_k}^{\mathcal{T}}(\cdot|s))\right] \leq \delta$                    $\triangleright$ Per Eq. (3)
        $w_{k+1} = \arg\max_{w} J_{\theta_{k+1},w_k}(\theta_{k+1},w)$
        s.t. $\mathbb{E}_{s\sim\rho_{\pi_{\theta_{k+1},w_k}^{\mathcal{T}}}}\left[D_{KL}(\pi_{\theta_{k+1},w_k}^{\mathcal{T}}(\cdot|s) \,\big|\big|\, \pi_{\theta_{k+1},w_{k+1}}^{\mathcal{T}}(\cdot|s))\right] \leq \delta$                    $\triangleright$ Per Eq. (4)
    **end for**
    $\pi_{\theta} \leftarrow$ extract $(\pi_{\theta^M,w^M}^{\mathcal{T}})$            $\triangleright$ Extract a single program based on architecture weights $w^M$
    $\theta_0 \leftarrow \theta_M$                $\triangleright$ The extracted policy inherits parameters from trained $\pi_{\theta^M,w^M}^{\mathcal{T}}$
    **for** $k = 0,1,2,\ldots,N$ **do**                        $\triangleright$ Train the chosen architecture
        $\theta_{k+1} = \arg\max_{\theta} J_{\theta_k}(\theta)$
        s.t. $\mathbb{E}_{s\sim\rho_{\pi_{\theta_k}}}\left[D_{KL}(\pi_{\theta_k}(\cdot|s) \,\big|\big|\, \pi_{\theta_{k+1}}(\cdot|s))\right] \leq \delta$
    **end for**
    **return** $\pi_{\theta_N}$

---

### A.2    THE PROGRAMMATIC HIGH-LEVEL PLANNING

The pseudocode of the synthesis algorithm for HRL policies is depicted in Algorithm 2. The goal is to synthesize a high-level control plan on the abstract model using abstract actions $\varphi_{\text{Act}}$ as constraints. An abstract action is specified together with a guard $\varphi_{\text{Guard}}$ that encodes the condition under which the action can occur. Suppose for now that we are searching over a high-level plan of a fixed number

$k$ of abstract action steps to reach the goal state $\varphi_{\text{Goal}}$ from the initial state $\varphi_{\text{Init}}$:

$$
\begin{aligned}
&\varphi_{\text{Init}}(x_0, y_0) \wedge \\
&\bigwedge_{i=1}^{k} \big( \varphi_{\text{Act}}(x_{i-1}, y_{i-1}, x_i, y_i, m_{i-1}, m_i) \wedge \varphi_{\text{Guard}}(x_{i-1}, y_{i-1}, x_i, y_i, m_{i-1}) \big) \wedge \qquad (6)\\
&\varphi_{\text{Goal}}(x_k, y_k)
\end{aligned}
$$

Constraint (6) can be solved by off-the-shelf constraint satisfiability solvers e.g. Z3 (de Moura & Bjørner, 2008). We incrementally search for longer and longer high-level plans, starting from $k = 1$ and increasing $k$ until the optimal high-level plan solution is found. A solved high-level plan is a sequence of sub-tasks: $\big( \beta(x_0, y_0), \pi_1, \beta(x_1, y_1) \big), \big( \beta(x_1, y_1), \pi_2, \beta(x_2, y_2) \big), \ldots, \big( \beta(x_{k-1}, y_{k-1}), \pi_k, \beta(x_k, y_k) \big)$ where we extract the intermediate high-level goals $x_1, y_1, \ldots, x_{k-1}, y_{k-1}$ from the solution of the statisfiability solver, and the concretization function $\beta$ converts abstract states back to concrete goals on the unabstracted environment for the agent to successively achieve. For example, the solved high-level plan for Ant ⊃-Maze navigation is depicted in Fig. 6c. For each sub-task $i$ that starts from a set of initial states $\beta(x_{i-1}, y_{i-1})$, we learn a new low-level policy $\pi_i$ that drives the agent to reach states in $\beta(x_i, y_i)$. The states of $\pi_i$ when reaching $\beta(x_i, y_i)$ serve as the initial states for training $\pi_{i+1}$. Notice that we do not directly use primitive functions as $\pi_i$, because oftentimes low-level policies need to compose primitive functions using different weights under different environment conditions to achieve a sub-task, e.g. Ant turning around at a corner with different poses and velocities while contacting walls in the process. We therefore learn each low-level policy $\pi_i$ as a programmatic ensemble policy composing primitive functions using the technique in Sec. 3 and distance-based rewards. This ensures that our HRL policies are interpretable at both high level and low level.

---

**Algorithm 2** Programmatic High-Level Planning

---

**Require:** Primitive Specifications $\varphi_{\text{Act}}$, abstract Action Guard $\varphi_{\text{Guard}}$, abstract initial $\varphi_{\text{init}}$ and goal $\varphi_{\text{goal}}$, time-step limit $T$ for the execution of each low-level policy
**Ensure:** A high-level plan $model$, and its low-level policies $\pi_0, \pi_1, \ldots, \pi_{k-1}$

    **for** $k = 1, 2, \ldots, M$ **do**                                       ▷ High-level policy planning
        $\phi_k \leftarrow \varphi_{\text{Init}}(x_0, y_0) \wedge \bigwedge_{i=1}^{k} \big( \varphi_{\text{Act}}(x_{i-1}, y_{i-1}, x_i, y_i) \wedge \varphi_{\text{Guard}}(x_{i-1}, y_{i-1}, x_i, y_i) \big)$
                                      $\wedge \varphi_{\text{Goal}}(x_k, y_k)$         ▷ Per Eq. (5)
        $model \leftarrow \text{solve}\ (\phi_k)$
        **if** $model \neq$ None **then**
            $\forall i.\ x_i,\ y_i = model[x_i],\ model[y_i]$         ▷ A high-level control plan is synthesized
            **Break**
        **else if** $k == M$ **then**
            **Return "Failed"**
        **end if**
    **end for**
    **for** $i = 0, 1, 2, \ldots, k - 1$ **do**                               ▷ Low-level policy learning
        $R_i \leftarrow \lambda s, a.\ - dist(s, \beta(x_i, y_i))$         ▷ Reward based on distance to sub goal $(x_i, y_i)$
        $\pi_i = \arg\max_{\pi_i} \mathbb{E}_{s_0 \sim (\pi_0, \ldots, \pi_{i-1}), (a_0, s_1 \cdots) \sim \pi_i} \left[ \sum_0^T \gamma^t \cdot R_i(s_t, a_t) \right]$
                                      ▷ Apply Algorithm 1 to learn a low-level programmatic policy $\pi_i$
                                      ▷ The initial state $s_0$ is sampled as the last state observed by $\pi_{i-1}$
    **end for**
    **return** $model, (\pi_0,\ \pi_1,\ \ldots,\ \pi_{k-1})$

---

# B MORE ABLATION STUDY RESULTS

Our first ablation study investigates the impact of compositionality on our algorithm. Fig. 8 shows the ablation study results on comparing different strategies, i.e., learning a programmatic ensemble policy, learning a programmatic affine policy, and learning a deep neural network policy using TRPO (Schulman et al., 2015). A programmatic affine policy is a monolithic program in our DSL (Fig. 1)

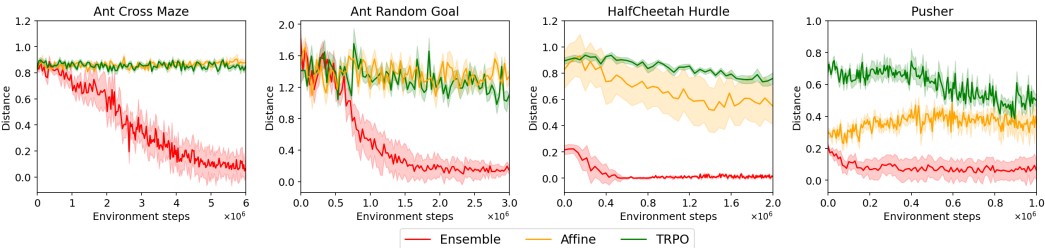

Figure 8: Ablation study on the importance of compositionality. Results are averaged over 10 random seeds. The $y$-axis records the agent's distance towards its goal. We normalize the distances to a final goal by the agent initial distance from the goal so values close to 1 or higher show failures.

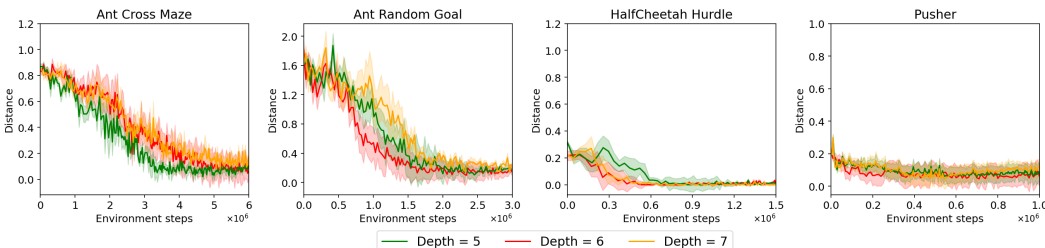

Figure 9: Ablation study on the impact of the depth bound of a program derivation tree. Results are averaged over 10 random seeds. The $y$-axis records the agent's distance towards its goal. We normalize the distances to a final goal by the agent initial distance from the goal so values close to 1 or higher show failures.

that switches back and forth between a set of *affine* controllers under different environment conditions. The result shows that our programmatic ensemble policies significantly outperform the other two representations in all group one environments, which highlights the merits of composition. Notice that in the HalfCheetah Hurdle and Pusher environments, the programmatic affine policies significantly outperform the neural network policies.

Our second ablation study investigates the impact of the depth bound of a program derivation tree on the performance of our algorithm. Fig. 9 shows the ablation study result about using different depth bounds on program derivation trees and compares the mean normalized final distances to goals of learned policies with standard deviations (averaged over 10 random seeds). Specifically, we compare program derivation trees with depths of 5, 6 and 7. The result demonstrates that the training performance is not sensitive to these depth bounds. The algorithm converges to architectures with similar performance. Table 2 additionally shows that the structures of the converged architectures are also similar. This suggests that if the user does not know how complex a programmatic policy should be a priori, the user can simply set a large tree-depth bound and let the learning algorithm assign the highest probability to the architecture that leads to a program maximizing cumulative RL reward.

## C   COMPARISON WITH ORACLE-GUIDED PROGRAMMATIC RL

One of our main contributions is direct programmatic policy search. Existing oracle-guided programmatic RL algorithms suffer from a nontrivial distillation gap as the distillation process can yield suboptimal programmatic policies whose reward performance is significantly worse than that of their oracles. This is because oracles (e.g. neural policies) and programs may reside in very different policy structure spaces. Due to the structural constraints, the program that best imitates an oracle is not necessarily a performant programmatic policy, or could even be much worse than the optimal program in the search space. To address this limitation, we propose programmatic RL solely based on reward signals without using any oracles. This allows the synthesis algorithm to search for optimal programs in the programmatic policy space. We compare our programmatic RL algorithm with oracle-based programmatic RL algorithms that learn policies in the form of programs

| Environment | Neural Oracles | Oracle-guided RL Programs | Decision Trees | Oracle-free RL Programs |
|---|---|---|---|---|
| Reacher | -5.01 | -5.79 | -4.79 | -5.05 |
| Walker2d | 4752 | 3672 | 1714 | 5178 |
| Hopper | 3634 | 1646 | 2995 | 3535 |
| HalfCheetah | 14627 | 3569 | 2810 | 10773 |
| Ant | 5786 | 4875 | 3504 | 5680 |
| Swimmer | 335 | 335 | 334 | 340 |
| BipedalWalker | 287 | 273 | 252 | 274 |
| Pendulum | -146 | -146 | -149 | -144 |

Table 3: Comparison between our programmatic RL algorithm with oracle-based programmatic RL algorithms that learn policies in the form of programs (using the same DSL as ours) (Verma et al., 2018) and decision trees (Bastani et al., 2018) on Mujoco/OpenAI environments. Neural oracles and our programmatic policies were trained using 3 million environment steps and we report the averaged final reward performance of three repeated experiments.

(using the same DSL as ours) (Verma et al., 2018) and decision trees (Bastani et al., 2018). In Table 3, we report results on Mujoco/OpenAI environments where programmatic policies invoke a set of *affine* controllers under different environmental conditions. To ensure a fair comparison, we relax the semantics of oracle-guided RL programs to be continuous as well but decision trees remain discrete. Neural oracles for Walker2d, Hopper, HalfCheetah, and Ant were trained with SAC (Haarnoja et al., 2018) using 3 million environment steps. Neural oracles for Reacher, Swimmer, BipedalWalker, and Pendulum were trained with TRPO (Schulman et al., 2015) using 3 million environment steps and we report the averaged final reward performance of three repeated experiments.

The reward performance of our oracle-free programmatic RL policies is comparable to neural network policies. Due to the distillation gap, the oracle-guided RL programs learned by the baselines perform worse than their oracles and our policies. We also tried the imitation-projected programmatic RL algorithm (Verma et al., 2019) that optimizes a programmatic policy by taking a mirror gradient descent in a policy space with a mix of neural and programmatic representations, which enables deep policy gradient approaches on programmatic policies. However, our own implementation of the algorithm in the mixed policy space does not lead to significant policy improvement compared with (Verma et al., 2018).

We further compare our algorithm with Verma et al. (2018) on the environments reported in Sec. 5. According to our results in Fig. 7a, these environments are best solved using policies that invoke a set of ensemble controllers. We report the mean normalized final distances to goals of learned policies with standard deviations. The neural oracles were trained by Compositional-SAC (Qureshi et al., 2020). Table 2 demonstrates that the oracle-guided RL programs ($\pi$-Oracle) are again suboptimal due to the distillation gap. Especially, they do not work well in environments with multiple goals. For example, we found that although on Ant Cross Maze the neural compositional-SAC policy in general works well, the success rate of reaching the goal (6, -6) is higher than reaching the other goals. The bias is amplified in the oracle-guided programmatic RL policy. When the goal is at (12, 0), the agent sometimes mistakenly goes somewhere near (6, -6).

The above two sets of experiments demonstrate that our oracle-free programmatic RL overcomes the suboptimality induced by oracle-guided programmatic RL.

# D   DSLs without Loops

Our loop-free programmatic policies can already produce repetitive behaviors when necessary. This is because policies are always executed in a feedback loop with RL environments. At each iteration, environment states encountered by repetitive behaviors would repeatedly activate or be matched to the same if-then-else conditions and therefore trigger alike actions. To support this argument, we evaluated our tool on two environments from Inala et al. (2020): Car and QuadPO. Both environments need policies that capture repeating behaviors. For Car, the goal is to drive a car out of a parking spot to an adjacent lane while avoiding collisions. For QuadPO, the goal is to maneuver a 2D quadcopter through an obstacle course by controlling its vertical acceleration. The training and test distributions are varied to evaluate whether policies can produce an arbitrary number of repetitions. For example,

| Environments | TRPO + NN | TRPO + Programmatic Affine Policy |
|:---:|:---:|:---:|
| Car | 82.6% | 100% |
| QuadPO | 69.6% | 85.6% |

Table 4: Comparison between our programmatic affine policies with neural network policies both learned based on a TRPO agent (Schulman et al., 2015). The training and test distributions are varied to evaluate whether policies can produce an arbitrary number of repetitions. We report the results on test distributions by measuring the fraction of rollouts (out of 500) that safely reach the goal.

on QuadPO, the obstacle course length is doubled during testing. Table 4 summarizes the results on test distributions by measuring the fraction of rollouts (out of 500) that safely reach the goal.

Our loop-free programmatic policies generalize better than neural policies on the test distributions. We did not compare our policies with the state-machine policies (Inala et al., 2020) as its implementation is not available.

Moreover, our tool indeed supports loops in a program that sequentially processes a history of environment states and actions. For example, our DSL allows a controller $C$ to be expanded as a discretized, multivariable PID controller in Fig. 1. In Appendix K.2, such a PID controller is formalized based on the higher-order combinator $fold$ that acts over a fixed-sized window on a history of observations. We could alternatively add $fold(f, h)$ directly to the DSL to search policies that combine the results of recursively processing each past observation in a history $h$ to build up a return action (the body of the combining operation $f$ can also be synthesized).

## E  ADDITIONAL RELATED WORK

**Compositional Reinforcement Learning.** To improve RL efficiency, a line of research that transfers past skills into new skills for solving new complex problems has emerged. Lee et al. (2019; 2020) learn high-level polices to select the previously trained primitive policies. The macro-action models (Vezhnevets et al., 2016) similarly learn a high-level planner to combine actions sequences from primitives. These methods activate only one policy at each timestep. Other works have explored simultaneously composing multiple task-agnostic primitive skills by learning to assign weights for action composition (Peng et al., 2019; Qureshi et al., 2020). Hierarchical reinforcement learning, such as option-critic (Bacon et al., 2017), is also a promising approach to solve complex tasks. However, option-critic is prone to inefficient task decomposition. Nachum et al. (2018) address this issue by automatically decomposing a complex task into subtasks and solving them by optimizing the subtask objectives. As opposed to these efforts, our method learns high-level and low-level policies that are both interpretable. Our technique can integrate any primitive skills diverse enough to complete a task and is orthogonal to automatic skill discovery methods e.g. (Eysenbach et al., 2019).

**Differentiable Architecture Search.** Neural architecture search has emerged as a promising approach to automate deep learning applications (Zoph & Le, 2017; Liu et al., 2018; Real et al., 2019). Particularly, our program architecture synthesis algorithm is inspired by DARTS (Liu et al., 2019b). This method uses a composition of softmaxes over all possible candidate operations between a fixed set of neural network nodes to relax the neural architecture search space in a super-network, and uses approximate gradient descent to iteratively train the weights and parameters of the super-network. It then selects the optimal architecture based on the weights. Various methods further improve architecture search efficiency, accuracy and applicability (Chen et al., 2019a; Guo et al., 2020; Zhao et al., 2021; Wang et al., 2021; Cui & Zhu, 2021; Jiang et al., 2019; Liu et al., 2019a). However, none of these methods has performed differentiable architecture search in a reinforcement learning context or considered leveraging symbolic logical abstractions for hierarchical architecture synthesis.

**Neural Program Synthesis.** Neural program synthesis involves learning neural networks to predict function distributions to guide a synthesizer e.g. DeepCoder (Balog et al., 2017), or generate a program autoregressively in an end-to-end fashion e.g. (Sun et al., 2018; Parisotto et al., 2017; Bunel et al., 2018) and RobustFill (Devlin et al., 2017). BUSTLE (Odena et al., 2021) uses bottom-up search that allows the model to prioritize and combine small programs that solve different subtasks. SKETCHADAPT (Nye et al., 2019) first generates a program sketch with holes, and then fills the holes using a conventional synthesizer. BAYOU (Murali et al., 2018) infers a different form of program sketches that abstract names and operations by their type. DreamCoder (Ellis et al., 2021)

| Models | Ant Cross Maze | Ant Random Goal | Pusher | Cheetah Hurdle |
|---|---|---|---|---|
| Program derivation tree | 0.0023s | 0.0023s | 0.0014s | 0.0014s |
| Programmatic policy (depth 6) | 0.0019s | 0.0019s | 0.0009s | 0.0009s |
| Bidirectional LSTM | 0.0034s | 0.0033s | 0.0022s | 0.0021s |

Table 5: The average running time of over 10000 random executions of program derivation trees, single programmatic policies, and Bidirectional LSTM models (Qureshi et al., 2020).

iteratively builds sketches using progressively more complicated primitives though a wake-sleep algorithm. Latent Programmer (Hong et al., 2021) considers two-level search for program synthesis, in which the synthesizer first generates a plan — a sequence of symbols, in other words discrete latent code from input/output examples by discrete autoencoders, that describes the desired program at a high level, and then generates the program in the target language. These methods avoid enumerating every possible program, which is prohibitively expensive for large program spaces.

There is also a line of work that deals with learning to process partial programs in addition to the specification. In execution-guided program synthesis, the model guides iterative expansions of a partial program until a matching one is found. Zohar & Wolf (2018) process intermediate values of a program using a neural network for search direction prioritization for a small, straight-line DSL. Ellis et al. (2019) allow values of whole and ground programs encountered during bottom-up search to be used to prioritize the search in an actor-critic framework. Similarly, Chen et al. (2019c) exploit intermediate values while synthesizing a program via a top-down fashion using a neural encoder-decoder model. Tian et al. (2019) exploit the differentiability of a rendering process to train a program generating policy and generalize it beyond the training distribution via gradient-based fine-tuning.

# F   IMPLEMENTATION DETAILS

In this section, we present implementation details including the optimization of $[\![E]\!](\cdot)$, primitive policies, symbolic inputs and reward structures of each environment, and hyperparameters of reinforcement learning algorithms.

## F.1   COMPUTATIONAL COMPLEXITY OF PROGRAM DERIVATION TREE EXECUTION

We compare the computational complexity of $[\![E]\!](\cdot)$ (where $d = 6$) with that of the deepest single program in $[\![E]\!](\cdot)$ and that of the Bidirectional LSTM-based model used by the Compositional SAC baseline (Qureshi et al., 2020) as follows. We report the average running time of over 10000 random executions of the three models in Table 5.

In our implementation, the run-time execution of a program derivation tree is optimized as follows. As depicted in Fig. 5, primitive policies are invoked multiple times by various programs embedded in a program derivation tree. Since the input to any primitive is always the current environment state, it suffices to call each primitive just once and use its result anywhere it is invoked in the tree policy. Table 5 shows that this simple optimization could enable us to run our algorithm as fast as the Compositional SAC baseline that does not perform any architecture search.

More importantly, we do not need to explicitly enumerate and execute all programs up to some depth for each state evaluated during an episode. Fig. 5 shows that we allow the exponential number of programs in a program derivation tree to share computation (similar to weight sharing in DARTS) — intermediate results computed by a shallower program can be reused by a deeper program.

For more complex DSLs with a large number $k$ of production rules, we could optimize the architecture search procedure by applying a strategy similar to Progressive DARTS (Chen et al., 2019b). This strategy gradually increases tree depth $d$ during search while dropping the search directions leading to lowest-weighted programs at the previous stage from the program derivation tree. Such a progressive procedure would allow our algorithm to deeply explore the architecture space even when $k$ is large, which is left for future work.

### F.2 PRIMITIVE POLICIES DETAILS

In all reinforcement learning tasks, we require multiple primitive skills policies to compose higher-level programs. Since primitive policies are obtained for completing dedicated tasks, e.g., navigating the ant agent to move up, or pushing down the object in the Pusher environment, it is unnecessary to include targets information in high-level tasks into their observation space during training. All primitive skills policies are obtained using SAC algorithm (Haarnoja et al., 2018) implemented in OpenAI Spinning Up RL framework (Achiam, 2018).

**Ant.** For standard-torque and low-torque ant, we train two sets of four basic primitive policies, i.e., *UP*, *DOWN*, *LEFT* and *RIGHT* for 1 million steps for each. The reward function to acquire above four primitives is defined as:

$$r_{ant} = c_v \cdot v_{direction} + c_h \cdot I(\text{IsHealthy}) - c_a \cdot \|a\|_2 - c_f \cdot \|f_{contact}\|_2 \tag{7}$$

where $v_{direction}$ is the velocity of given direction, $I(\text{IsHealthy})$ is a Boolean function that determines the health condition of the agent, action $a$ is a 8-dimension vector, and $f_{contact}$ is a vector that encodes the contact force of the agent. In order to train *UP* and *DOWN* primitives, $v_{direction} = \pm v_x$ should track velocities on $\pm x$-axis; similarly, to train *LEFT* and *RIGHT* policies, we should set $v_{direction} = \pm v_y$ to keep track of velocities along $\pm y$-axis. Coefficients $c_v$, $c_h$, $c_a$ and $c_f$ are set to be 1, 1, 0.5, $5 \times 10^{-4}$, respectively.

**Pusher.** The Pusher environment requires two skills: *PUSH-DOWN* and *PUSH-LEFT* used to manipulate the object to move down and left, and both of them are trained for 0.5 million steps. The reward function to acquire these two primitives is defined as:

$$r_{pusher} = c_v \cdot v_{direction} - c_a \cdot \|a\|_2 \tag{8}$$

where $v_{direction}$ denotes the object's velocity projected on *DOWN* (negative $y$-axis) or *LEFT* (positive $x$-axis), and $a$ is the action. Coefficients $c_v$ and $c_a$ are set to be 1 and 0.1, respectively.

**HalfCheetah.** In HalfCheetah Hurdle environment, two simple policies *JUMP* and *FORWARD* are trained for 0.5 million steps, and they are sufficient to compose the program and solve the task. The reward function can be defined as:

$$r_{hc} = c_{v_f} \cdot v_{forward} + c_{v_j} \cdot v_{jump} - c_a \cdot \|a\|_2 \tag{9}$$

where $v_{forward}$ and $v_{jump}$ are velocities projected on forward ($x$-axis) and jump ($y$-axis) directions, and $a$ is the action. In order to train *FORWARD* policy, coefficients $c_{v_f}$, $c_{v_j}$ are set to be 1 and 0; for policy *JUMP*, we set coefficients $c_{v_f} = 0.05$ and $c_{v_j} = 1$, respectively. Coefficient $c_a$ is the control cost coefficient which set to be 0.1.

Note that for Ant and HalfCheetah Hurdle environment, primitive policies are trained in a blank environment without any obstacle so that they can move freely.

### F.3 PROGRAM INPUTS

For low-torque ant, we consider its current position $x, y$, distance from current position to the starting point $\|x, y\|_2$, and the angle between current direction to the x-axis $arctan\frac{y}{x}$ as symbolic inputs. For standard-torque ant, besides the above inputs, we also take the goal position $\mathcal{G}_x, \mathcal{G}_y$ for navigation, since goals in environments which utilize the standard ant are randomly sampled. In the HalfCheetah Hurdle environment, the position of the next hurdle $\mathcal{H}_{next}$ and the distance from HalfCheetah's back foot to next hurdle $|x_{back} - \mathcal{H}_{next}|$ are considered. For the simple pusher task, we use distance from origin to the object $\|x_{obj}, y_{obj}\|_2$ and to the robotic arm $\|x_{arm}, y_{arm}\|_2$ as symbolic inputs.

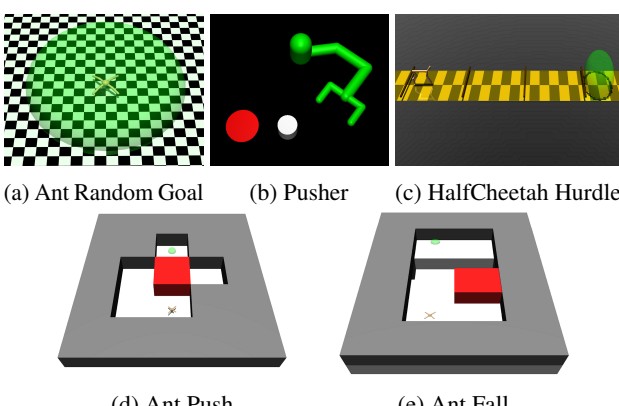

(a) Ant Random Goal    (b) Pusher    (c) HalfCheetah Hurdle

(d) Ant Push      (e) Ant Fall

Figure 10: Navigation and manipulation benchmarks that require agents to reach or move the object to the given targets (represented by a red circle for pusher and by green spheres for the rest). Additionally, Ant Cross Maze and Ant ⊃-Maze are depicted in Fig. 2 and Fig. 6a respectively. In all ant environments, the definitions of up, down, left, right, and forward are moving towards positive/negative x/y axes directions, respectively.

## F.4 ENVIRONMENT DETAILS

### F.4.1 ANT ENVIRONMENTS

The standard-torque Ant environments include **Ant Cross Maze** and **Ant Random Goal** environments. In these environments, a MuJoCo quadruped ant with 150 units torque limit is required to start from origin and reach the random given targets. The radius of the goals are 1 and 0.25, respectively. Specifically, the goal of Ant Cross Maze is randomly sampled from one of three positions: (6, 6), (12, 0) and (6, -6) at each time; and the goals in Ant Random Goal is a random position within a circle with radius of 5. In ant environments, the reward function is defined as:

$$r_{ant\_rl} = c_{distance} \cdot (c_d - \|p_{ant} - p_{goal}\|_2) + c_h \cdot I(\text{IsHealthy}) \\ - c_a \cdot \|a\|_2 - c_f \cdot \|f_{contact}\|_2 \tag{10}$$

where $p_{ant} = (x_{ant}, y_{ant})$ and $p_{goal} = (x_{goal}, y_{goal})$ are positions of ant and the given targets, $I(\text{IsHealthy})$ is a Boolean function that determines the health condition of the agent, action $a$ is a vector, and $f_{contact}$ is a vector denotes the contact force. Coefficients $c_{distance}, c_d, c_h, c_a$ and $c_f$ are set to be 1, 5, 0.05, 0.01 and 0.001, respectively.

Additionally, we consider three challenging low-torque MuJoCo Ant environments with weak distance-based reward signals. Compared to standard-torque, the low-torque ant has the force range [-30, 30]. Since smaller force is exerted on joints, the ant moves slower and suitable for intricate HRL tasks. Specifically, we normalize the distances to a fixed final goal by the agent initial distance from the goal so values close to 1 or higher show failure. Greedily following this kind of reward signals does not lead to solving the problems as these environments are in complex shapes.

In the **Ant ⊃-Maze** environment, we place the Ant in a ⊃-shaped maze for a navigation task. The agent is trained to reach the farthest end of the maze located at (0, 19). In the **Ant Push** environment, the moveable block is at (0, 8), and the goal is at (0, 19). In the **Ant Fall** environment, the movable block is at (8, 8) at the same elevation as Ant. Their is a rift in the region [-4, 12] × [12, 20]. To reach the target on the other side of the rift, the Ant must push the block down into the rift, and then step on it to get to the goal position. For all three environments, the low-torque Ant starts from the origin, and the radius of goals are set to be 1.

### F.4.2 PUSHER ENVIRONMENT

In the **Pusher** environment, the initial position of the object is randomly sampled from the region [0.3, 1] × [-1, -0.4]. In order to successfully move the object to the given target centered at (-1, 0) with radius of 0.17 using the robotic arm, the distance from arm to object, as well as object to goal should be tracked. The reward function for Pusher environment is defined as:

$$r_{pusher2d} = -c_{reach} \cdot \|p_{arm} - p_{obj}\|_2 - c_{move} \cdot \|p_{obj} - p_{goal}\|_2 - c_a \cdot \|a\|_2 \tag{11}$$

where $p_{arm} = (x_{arm}, y_{arm})$, $p_{obj} = (x_{obj}, y_{obj})$, and $p_{goal} = (x_{goal}, y_{goal})$ are positions of the robotic arm, object, and the goal, respectively. Coefficient $c_{reach}$, $c_{move}$, $c_a$ are set to be 0.5, 1, and 0.1, respectively.

### F.4.3 HALFCHEETAH HURDLE ENVIRONMENT

In the **HalfCheetah Hurdle** environment, a MuJoCo halfcheetah is required to start from origin and jump over three hurdles to reach the target with radius of 1. The reward function is defined as:

$$r_{hurdle} = - c_{distance} \cdot |x_{goal} - x| + c_{hurdle} \cdot count(x) \\ + c_{goal} \cdot I(\text{IsReached}) + c_{run} \cdot v_x + c_{jump} \cdot v_y - c_{collision} \cdot I(\text{IsCollided}) \tag{12}$$

where $x$ is cheetah's current position, $x_{goal}$ is target location, $count(x)$ function returns the number of hurdles has been passed, $I(\text{IsReached})$ is a Boolean function that determines if the cheetah reach the target, $v_x$ and $v_y$ are the projected velocities on $x$-axis and $y$-axis, and $I(\text{IsCollided})$ is a Boolean function that determines if the cheetah is in collision. Coefficient $c_{distance}$, $c_{hurdle}$, $c_{goal}$, $c_{run}$, $c_{jump}$ and $c_{collision}$ are set to be 0.1, 1, 1000, 1, 0.3 and 2, respectively.

### F.5 HYPERPARAMETERS

Following hyperparameters are used to train primitive policies with SAC (Haarnoja et al., 2018) algorithm.

- Discount factor $\gamma = 0.99$.
- SGD optimizer; actor learning rate 0.001; critic learning rate 0.001.
- Mini-batch size $n = 100$.
- Replay buffer of size 100000.
- Soft update targets $\tau = 0.005$.
- Target update interval and gradient step are set to be 1.

Following hyperparameters are used to train $\pi$-**PRL** programs for solving tasks in group one environments with TRPO (Schulman et al., 2015) algorithm.

- Discount factor $\gamma = 0.99$.
- Number of trajectories per epoch $N = 50$.
- Maximum search depth of program derivation graph $D_g = 6$.
- KL-Divergence limit $\delta = 0.01$.
- GAE $\lambda = 0.97$.
- Gumbel-Softmax Temperature $T = 0.25$.

Following hyperparameters are used to train $\pi$-**HPRL** programs for solving tasks in group two environments with TRPO algorithm.

- Discount factor $\gamma = 0.995$.
- Number of trajectories per epoch $N = 100$.
- KL-Divergence limit $\delta = 0.01$.
- GAE $\lambda = 0.97$.
- Gumbel-Softmax Temperature $T = 0.25$.

Table 6 summarizes the neural network architectures used in reinforcement learning algorithms.

| Method | Network Structure | Hidden Units |
|---|---|---|
| HIRO | High-level Policy: Three layer feed forward network | 300 |
| | Low-level Policy: Three layer feed forward network | 300 |
| Composition-SAC | Encoder: Bidirectional RNN with LSTMs | 128 |
| | Decoder: Single layer feed forward network | 128 |
| | Attention:$W_f, W_b, W_d \in \mathbb{R}^{d \times d}, W \in \mathbb{R}^d$ | 128 |
| TRPO/PPO policy | Two layer feed forward network | 256 |
| SAC primitive policy | Two layer feed forward network | 256 |

Table 6: Network Structures.

## G IMPLEMENTATION DETAILS OF PROGRAMMATIC HIGH-LEVEL PLANNING

### G.1 SPECIFICATIONS FOR HIGH-LEVEL PLANNING

In our experiment, we have applied programmatic high-level planning in Sec. 4 to 3 challenging environments, namely Ant ⊃-Maze (Fig. 6a), Ant Push (Fig. 10d), and Ant Fall (Fig. 10e), that are commonly used to evaluate the effectiveness of hierarchical reinforcement learning. Sec. 4 shows how to perform high-level planning based on the ensemble policy DSL and uses Ant ⊃-Maze as an example. In this section, we apply programmatic high-level planning to solve Ant Push and Ant Fall.

Recall that our method builds an abstract high-level model based on state abstractions provided by a domain expert, similar to previous works (Abel et al., 2020; Winder et al., 2020; Jothimurugan et al., 2021). For example, consider the Ant navigation task in a ⊃-shaped maze from Nachum et al. (2018) in Fig. 6a. The abstract states of the task are collected on an abstract 2-D gridworld $m \in \mathbb{R}^{N \times N}$ in Fig. 6b. An abstract grid $(x, y) \in N \times N$ subsumes concrete positions with a known scale where $m[x, y] = 1$ indicates walls and $m[x, y] = 0$ indicates nagivable spaces. The domain expert can then specify a pair of abstract initial state $\varphi_{\text{Init}}$ (yellow) and goal state $\varphi_{\text{Goal}}$ (green). By abstracting away details of the low-level dynamics such as Ant orientation and velocity, a high-level policy over abstract states and actions can efficiently plan over much longer time horizons.

For Ant Push and Ant Fall, we reuse the same state abstraction as above. We additionally use $m[x, y] = \text{Move}$ to specify that there is a movable block at $(x, y)$. The Ant needs to push away the movable block to reach the goal behind it in Ant Push or push it into a rift and then utilize the block as a bridge to get close to the goal on the other side of the rift.

Other than state abstraction, for model-based planning, we also abstract over actions to ensure a finite high-level model. We directly use action abstractions as provided primitive functions therein the ensemble policy DSL because they abstract low-level agent actions to an abstract space of skills. Our algorithm then leverages abstract action (primitive function) specifications to synthesize a high-level control plan on the abstract high-level model.

Compared to existing model-based HRL approaches e.g. (Jothimurugan et al., 2021), our algorithm uses abstract actions as specified rather than learned. In the following, we show that with task-agnostic abstract actions (primitive functions), providing these specifications is trivial even for complex high-level planning tasks.

For the Ant Push and Ant Fall environments, the Ant primitives move an Ant up $\pi_{\text{up}}$, down $\pi_{\text{down}}$, left $\pi_{\text{left}}$, and right $\pi_{\text{right}}$. Therefore, the abstract actions move an Ant horizontally, vertically, or diagonally (via composition) one step on the abstract model:

$$\varphi_{\text{Act\_Ant\_Push}}(x_{i-1}, y_{i-1}, x_i, y_i, m_{i-1}, m_i) \equiv$$
$$\exists dx_i, dy_i.\ x_i = x_{i-1} + dx_i\ \wedge\ y_i = y_{i-1} + dy_i\ \wedge -1 \leq dx_i,\ dy_i \leq 1\ \wedge$$
$$\forall p, q.\ m_i[p][q] = \begin{cases} 0 & m_{i-1}[x_i][y_i] = \text{Move} \wedge p = x_i \wedge q = y_i \\ \text{MOVE} & m_{i-1}[x_i][y_i] = \text{Move} \wedge p = x_i + dx_i \wedge q = y_i + dy_i \\ m_{i-1}[p][q] & \text{otherwise} \end{cases}$$

Compared to the $\varphi_{\text{Act}}$ definition in Sec. 4, we add two new parameters $m_{i-1}$ and $m_i$ to the new definition above. This is because an agent action may change its environment. Here $m_{i-1}$ refers to the old environment before an agent action, and $m_i$ refers to the new environment after the agent

| Environment | $\pi$-HRPL | | HIRO | | A-AVI | |
|---|---|---|---|---|---|---|
| Ant $\supset$-Maze | **0.06±0.02** | **84%±6%** | 0.27±0.04 | 82%±4% | 0.18±0.02 | 77%±8% |
| Ant Push | **0.07±0.02** | **94%±8%** | 0.19±0.08 | 46%±8% | 0.40±0.08 | 42%±12% |
| Ant Fall | **0.08±0.04** | **65%±5%** | 0.42±0.08 | 37%±15% | 0.31±0.03 | 52%±5% |

Table 7: Performance comparison for group two environments averaged over 5 random seeds. The mean normalized final distances to goals and success rates with their standard deviations are reported.

action. For example, for Ant Push, if a movable block is pushed by the Ant, the state abstraction should be updated accordingly reflecting the fact that the space after the movable block in $m_{i-1}$ holds the block after the push in the new environment $m_i$.

We require that an abstract action be specified together with a guard $\varphi_{\text{Guard}}$ that encodes the condition under which the action can occur. For Ant Push, the Ant should not walk into walls and the path from an old position to a new position should be available. Importantly, if in the high-level plan the Ant pushes the movable block, there should exist space behind the block that allows for the push in the old environment $m_{i-1}$ before the action. Besides, a movable block can only be moved vertically or horizontally in the environment.

$$\varphi_{\text{Guard\_Ant\_Push}}(x_{i-1}, y_{i-1}, x_i, y_i, m_{i-1}) \equiv$$
$$m_{i-1}[x_i][y_i] \neq 1 \wedge \left(m_{i-1}[x_{i-1}][y_i] = 0 \vee m_{i-1}[x_i][y_{i-1}] = 0\right) \wedge$$
$$m_{i-1}[x_i][y_i] = \text{Move} \Rightarrow m_{i-1}[x_i + (x_i - x_{i-1})][y_i + (y_i - y_{i-1})] = 0$$
$$m_{i-1}[x_i][y_i] = \text{Move} \Rightarrow x_i = x_{i-1} \vee y_i = y_{i-1}$$

The abstract action specification for Ant Fall can be defined similarly as follows:

$$\varphi_{\text{Act\_Ant\_Fall}}(x_{i-1}, y_{i-1}, x_i, y_i, m_{i-1}, m_i) \equiv$$
$$x_i = x_{i-1} + dx_i \wedge y_i = y_{i-1} + dy_i \wedge -1 \leq dx_i, \ dy_i \leq 1 \wedge$$
$$\forall p, q. \ m_i[p][q] = \begin{cases} 0 & m_{i-1}[x_i][y_i] = \text{Move} \wedge p = x_i \wedge q = y_i \\ 0 & m_{i-1}[x_i][y_i] = \text{Move} \wedge m_{i-1}[p][q] = \text{Rift} \wedge \\ & \qquad p = x_i + dx_i \wedge q = y_i + dy_i \\ \text{Move} & m_{i-1}[x_i][y_i] = \text{Move} \wedge m_{i-1}[p][q] \neq \text{Rift} \wedge \\ & \qquad p = x_i + dx_i \wedge q = y_i + dy_i \\ m_{i-1}[p][q] & \text{otherwise} \end{cases}$$

Notice that $\varphi_{\text{Act\_Ant\_Fall}}$ is almost the same as $\varphi_{\text{Act\_Ant\_Push}}$ except that we specify that if the Ant pushes a movable block into a rift, the rift becomes a bridge for the Ant to bypass.

$$\varphi_{\text{Guard\_Ant\_Fall}}(x_{i-1}, y_{i-1}, x_i, y_i, m_{i-1}) \equiv$$
$$m_{i-1}[x_i][y_i] \neq 1 \wedge \left(m_{i-1}[x_{i-1}][y_i] = 0 \vee m_{i-1}[x_i][y_{i-1}] = 0\right) \wedge$$
$$m_{i-1}[x_i][y_i] = \text{Move} \Rightarrow m_{i-1}[x_i + (x_i - x_{i-1})][y_i + (y_i - y_{i-1})] = 0 \vee$$
$$m_{i-1}[x_i + (x_i - x_{i-1})][y_i + (y_i - y_{i-1})] = \text{Rift}$$
$$m_{i-1}[x_i][y_i] = \text{Move} \Rightarrow x_i = x_{i-1} \vee y_i = y_{i-1}$$

Similarly, $\varphi_{\text{Guard\_Ant\_Fall}}$ is almost the same as $\varphi_{\text{Guard\_Ant\_Push}}$ except that we additionally specify that the Ant can push a movable block either if there is an empty space or a rift behind the block in the old environment before the action.

## G.2 DETAILED RESULTS ON HIERARCHICAL RL BENCHMARKS

We show the success rates of applying our learning algorithm (Sec. 4), HIRO (Nachum et al., 2018), and A-AVI (Jothimurugan et al., 2021) in Table 7.

It can be seen that $\pi$-HRPL programs significantly outperform both HIRO and A-AVI. For example, on the challenging Ant Push environment, our $\pi$-HRPL policy achieves 94% success rate of goal reaching far beyond that of HIRO and A-AVI.

## H PROGRAM INTERPRETABILITY: ANT CROSS MAZE

In this section, we take one step ahead and show how an Ant Cross Maze program $\mathcal{P}_{cross}$ can be interpreted by using both branch activation and also policy strength graphs.

### H.1 BRANCH ACTIVATION

In Sec.2, we demonstrate multiple branch activation graphs given different two targets: (6, -6) and (12, 0), and we analyze how they can be explained. Furthermore, When the goal is set to (6, 6), in most areas of the maze, branch one of the $\mathcal{P}_{cross}$ is activated (shown in Fig. 11a) and the agent would majorly follow the instruction provided by the branch one. From Fig.3, we can conclude that the semantics of this branch is to navigate the agent to go *UP*, since the parameter associated with *UP* primitive policies dominates this branch. Similarly, Fig.11b shows that int most upper-half of the given area, branch two is activated and it would enforce the agent to *go RIGHT*.

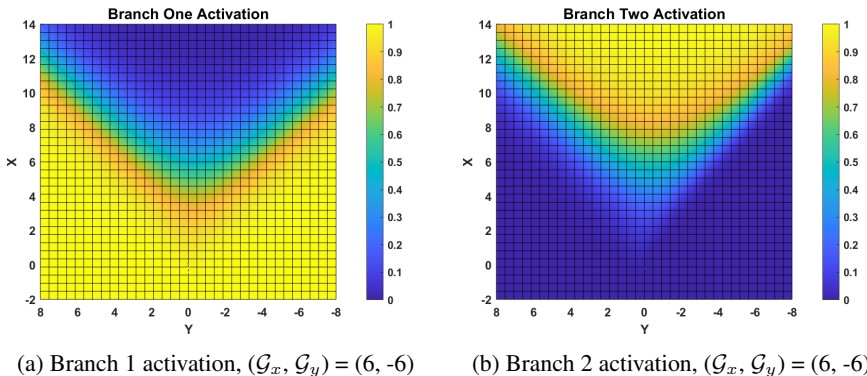

(a) Branch 1 activation, $(\mathcal{G}_x, \mathcal{G}_y) = (6, -6)$      (b) Branch 2 activation, $(\mathcal{G}_x, \mathcal{G}_y) = (6, -6)$

Figure 11: Branch activation as functions of Ant position $(x, y)$ for program $\mathcal{P}_{cross}$.

### H.2 POLICY STRENGTH

Unlike a branch activation graph which informs us how a branch is activated, a policy strength graph demonstrates how a specific primitive skill policy is invoked without considering the semantic of any branch, so it is suitable for final analysis and interpretation.

Fig.12a and 12b demonstrate the *UP* policy strength of program $\mathcal{P}_{cross}$ given target locations (12, 0) and (6, -6). Presumably, if the agent start from origin (0, 0) and the target is (12, 0), it should move straight up and invoke *UP* policy. From the Fig.12a we know that the behavior of the program is consistent with our expectation: invoking *UP* policy throughout its trajectory. Furthermore, if we change the target position to (6, -6), behavior of how the program invokes *UP* policy (see Fig.12b) changes accordingly: the strength of *UP* policy is at a high level in the beginning of the trajectory, then decreases rapidly to prepare for a right turn. Therefore, the effect of $\mathcal{P}_{cross}$ shows in Fig.12 can be summarized as: *the strength of UP policy could be adjusted according to different target locations, yet it stays at a high level at the beginning of the trajectory.*

The goals shown in Fig.13a and 13b are symmetric. After reaching the central area of the maze using *UP* policy, the program should navigate the ant to make a left or right turn by calling *LEFT* or *RIGHT* primitive policies, respectively. If the goal is located at (6, 6) (see Fig.13a), the *LEFT* policy strength is high after the agent reaches the center of the maze, so it can be pushed and move towards left. When the goal is assigned to (6, -6) on the right side of the maze (see Fig.13b), the *RIGHT* policy is invoked at a high level after the ant reach the center of the maze, so that the agent will be pushed to move right. Therefore, the effect of $\mathcal{P}_{cross}$ shows in Fig.13 can be summarized as: *the LEFT or RIGHT polices strength is at a high level when the ant reaches the center area of maze so it can be navigated to make a left or right turn to reach the targets.*

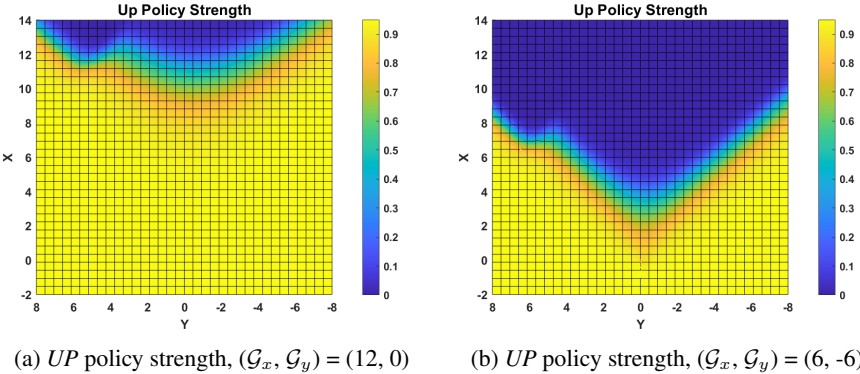

(a) *UP* policy strength, $(\mathcal{G}_x, \mathcal{G}_y) = (12, 0)$     (b) *UP* policy strength, $(\mathcal{G}_x, \mathcal{G}_y) = (6, -6)$

Figure 12: *UP* policy strength of a Ant Cross Maze program $\mathcal{P}_{cross}$

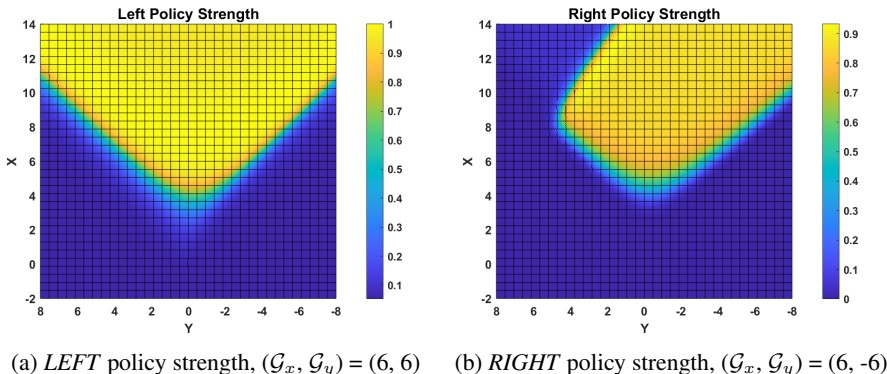

(a) *LEFT* policy strength, $(\mathcal{G}_x, \mathcal{G}_y) = (6, 6)$     (b) *RIGHT* policy strength, $(\mathcal{G}_x, \mathcal{G}_y) = (6, -6)$

Figure 13: *LEFT* and *RIGHT* policy strength of a Ant Cross Maze program $\mathcal{P}_{cross}$.

# I PROGRAM INTERPRETABILITY: MORE EXAMPLES

In this section, we demonstrate more examples of programs and their branch activation and policy strength graphs for interpretation.

## I.1 ANT RANDOM GOAL PROGRAM

Fig.14 demonstrates the expression of the program $\mathcal{P}_{random}$ for solving Ant Random Goal environment. The pre-defined function input $\mathcal{X}$ takes symbolic states include current position $x, y$ along with the randomly sampled target location $\mathcal{G}_x, \mathcal{G}_y$. Also, a vector of distribution of weights of primitive policies can be found on each terminal branch.

Fig.15 shows the branch activation of the first three branches. The figures clearly demonstrate that when the goal is set to be $(\mathcal{G}_x, \mathcal{G}_y) = (-3, -3)$, roughly only branch one is activated in the needed areas, so the behavior of the agent is dominantly determined by the semantic of branch one. Since the second and fourth values in the policy distribution vector in branch one nearly equally dominate the branch, the expected behavior of this branch can be summarized as: *go RIGHT and go DOWN at the same time.*

Fig.16a and 16b demonstrate the *DOWN* and *RIGHT* policy strength of a program $\mathcal{P}_{random}$ given origin $(0, 0)$ and target location $(-3, -3)$, respectively. In the figures, the x-axis and y-axis reflect the current position of the agent. The output of $\mathcal{P}_{random}$ is a vector that contains strength of each primitive skill policy and we are interested in the strength *DOWN* and *RIGHT* primitive policies in this case. Presumably, if the goal position is $(-3, -3)$ and the ant starts from $(0, 0)$, the simplest path for an ant is to move diagonally along with southeast direction, and surely this requires *DOWN* and *RIGHT* primitive policies, and each of them is likely to contribute equally. From Fig.16 we know that with this program $\mathcal{P}_{random}$, when the agent moves southeast, the contribution of *DOWN* and

if $(\theta_{1_c} + \theta_1^T \cdot \mathcal{X} > 0)$
    then $(0\% \cdot \pi_{\text{UP}}(s) + 51\% \cdot \pi_{\text{DOWN}}(s) + 0\% \cdot \pi_{\text{LEFT}}(s) + 49\% \cdot \pi_{\text{RIGHT}}(s))$
    else if $(\theta_{2_c} + \theta_2^T \cdot \mathcal{X} > 0)$
        then $(3\% \cdot \pi_{\text{UP}}(s) + 0\% \cdot \pi_{\text{DOWN}}(s) + 97\% \cdot \pi_{\text{LEFT}}(s) + 0\% \cdot \pi_{\text{RIGHT}}(s))$
        else if $(\theta_{3_c} + \theta_3^T \cdot \mathcal{X} > 0)$
            then $(82\% \cdot \pi_{\text{UP}}(s) + 15\% \cdot \pi_{\text{DOWN}}(s) + 0\% \cdot \pi_{\text{LEFT}}(s) + 3\% \cdot \pi_{\text{RIGHT}}(s))$
            else $(78\% \cdot \pi_{\text{UP}}(s) + 0\% \cdot \pi_{\text{DOWN}}(s) + 0\% \cdot \pi_{\text{LEFT}}(s) + 22\% \cdot \pi_{\text{RIGHT}}(s))$

$\mathcal{X} = [\, x, \, y, \, \mathcal{G}_x, \, \mathcal{G}_y, arctan\frac{y}{x}, \|x, y\|_2 \,]$
$\theta_1 = [\, 0.899, 0.397, -0.893, -0.394, -0.019, 0.080 \,], \; \theta_{1_c} = -2.050$
$\theta_2 = [\, 0.222, -1.679, -0.117, 1.468, -0.107, 0.297 \,], \; \theta_{2_c} = -3.798$
$\theta_3 = [\, -0.895, 0.196, -1.290, 2.423, -2.667, 2.378 \,], \; \theta_{3_c} = -2.226$

Figure 14: An Ant Random Goal program $\mathcal{P}_{random}$ powered by four primitive skill policies: $\pi_{\text{UP}}(s), \pi_{\text{DOWN}}(s), \pi_{\text{LEFT}}(s)$ and $\pi_{\text{RIGHT}}(s)$. Program input $\mathcal{X}$ includes current Ant position $x, y$, target goal $\mathcal{G}_x, \mathcal{G}_y$, $arctan\frac{y}{x}$, and $\|x, y\|_2$.

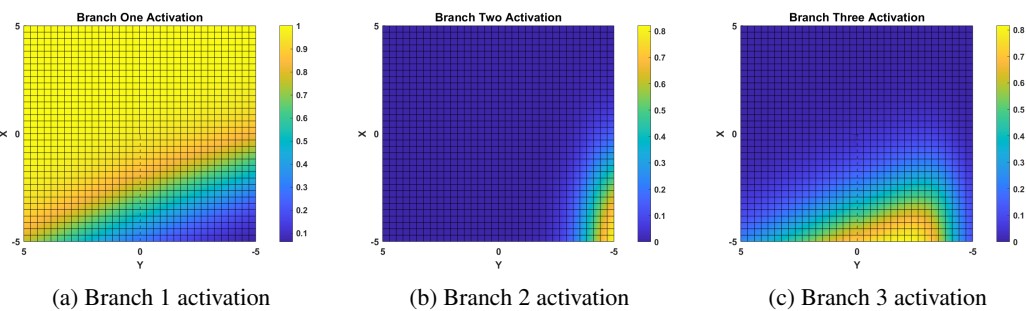

(a) Branch 1 activation      (b) Branch 2 activation      (c) Branch 3 activation

Figure 15: Branch activation of an Ant Random Goal program $\mathcal{P}_{random}$, $(\mathcal{G}_x, \mathcal{G}_y)$ = (-3, -3).

*RIGHT* are roughly the same, and this is consistent with our expectation. Therefore, the effect of this program can be summarized as: *if the agent is required to move southeast from (0, 0) to (-3, -3), assign DOWN and RIGHT equally at roughly 50 percent of strength.*

### I.2    PUSHER PROGRAM

Fig.17 demonstrates a program $\mathcal{P}_{pusher}$ with three branches for solving Pusher environment. The program input $\mathcal{X}$ takes symbolic states $x_{obj}$, $y_{obj}$, $x_{arm}$, $y_{arm}$ as inputs, then computes $\|x_{obj}, y_{obj}\|_2$, $\|x_{arm}, y_{arm}\|_2$ to decide which branch of weight cell to take.

From Fig.18a we can conclude that, only the first branch is activated in the whole possible situations. Such activation values for branch 1 demonstrate that the possibility of cutting other branches in the program without hurting the performance of the graph too much. Also, it suggests that a smaller value of maximum depth of program derivation graph is sufficient for searching a program to solve this task. Also, the distribution of primitive policies in this branch shows that: *the agent needs more PUSH-DOWN policy than PUSH-LEFT policy to move the object to the target location.*

Furthermore, Fig.18 demonstrates *PUSH-DOWN* and *PUSH-LEFT* policy strength of the program $\mathcal{P}_{pusher}$. Unlike Ant Cross Maze or Ant Random Goal environment, the goal of pusher environment is at a fixed location. From Fig.18b we can conclude that: *when the object is faraway from the origin, i.e., close to the goal, PUSH-DOWN policy strength is at a relatively lower level, compared to at a higher level when the object is far from the goal position.* Also, we can summarize the effect of how the program invoke *PUSH-LEFT* primitive policy from Fig.18c: *when the object is too close to the origin, the PUSH-LEFT policy strength is at a lower level, and if the object is close to the goal, the PUSH-LEFT policy strength is at a higher level.* In general, the above behavior of *PUSH-LEFT* can be explained if we consider velocities in both x and y axes: if we want to acquire a policy which

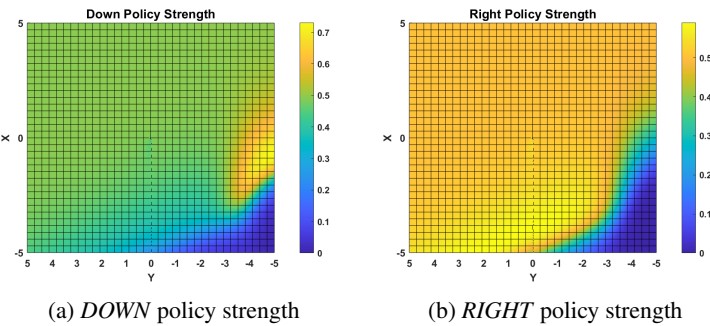

(a) *DOWN* policy strength     (b) *RIGHT* policy strength

Figure 16: *DOWN* and *RIGHT* policy strength of program $\mathcal{P}_{random}$, $(\mathcal{G}_x, \mathcal{G}_y)$ = (-3, -3).

$$
\begin{aligned}
&\textbf{if } (\theta_{1_c} + \theta_1^T \cdot \mathcal{X} > 0) \\
&\quad \textbf{then } (\ 74\% \cdot \pi_{\text{PUSH-DOWN}}(s) + 26\% \cdot \pi_{\text{PUSH-LEFT}}(s)\ ) \\
&\quad \textbf{else if } (\theta_{2_c} + \theta_2^T \cdot \mathcal{X} > 0) \\
&\quad\quad \textbf{then } (\ 99\% \cdot \pi_{\text{PUSH-DOWN}}(s) + 1\% \cdot \pi_{\text{PUSH-LEFT}}(s)\ ) \\
&\quad\quad \textbf{else } (\ 99\% \cdot \pi_{\text{PUSH-DOWN}}(s) + 1\% \cdot \pi_{\text{PUSH-LEFT}}(s)\ ) \\
\\
&\mathcal{X} = [\ \|x_{obj}, y_{obj}\|_2, \|x_{arm}, y_{arm}\|_2\ ] \\
&\theta_1 = [\ -1.370, 0.621\ ], \theta_{1_c} = 5.041 \\
&\theta_2 = [\ -1.252, -8.101\ ], \theta_{2_c} = -3.426
\end{aligned}
$$

Figure 17: A Pusher program $\mathcal{P}_{pusher}$ powered by two primitive skill policies: $\pi_{\text{PUSH-DOWN}}(s)$ and $\pi_{\text{PUSH-LEFT}}(s)$. Program input $\mathcal{X}$ includes norms over $x_{obj}, y_{obj}$ and $x_{arm}, y_{arm}$.

push the object at a constant tangential velocity all the way to the goal, we may need to adjust the distribution of *PUSH-LEFT* and *PUSH-DOWN* in order to get a reasonable combination of velocity in both x and y axes.

### I.3 HALFCHEETAH HURDLE PROGRAM

According to our experiments when applying $\pi$-**PRL** algorithm to HalfCheetah Hurdle environment, the program derivation graph is inclined to select a program with few branches. In fact, a simple HalfCheetah Hurdle program $\mathcal{P}_{hc}$ with only one branch (shown in Fig.19) is able to solve the task with high performance and success rate.

An example of such program can be written as $95\% \cdot \pi_{\text{FORWARD}}(s) + 5\% \cdot \pi_{\text{JUMP}}(s)$. In this simple program, the input $\mathcal{X}$ is not even used, this means the agent invokes each primitive policy at a fixed rate. The success of such a simple combination of two primitive skill policies tells us, if a program is powered by appropriate low-level primitive policies, we may solve a RL task in a much simpler way.

## J POLICY GENERALIZABILITY

In our experience, the high interpretability of learned programmatic policies leads to strong generalizability of the policies to novel environments.

### J.1 ANT NO MAZE

We consider applying the programmatic policy $\mathcal{P}_{cross}$ for Ant Cross Maze to an environment with randomly sampled target positions far beyond the three possible goals in the original environment. Our goal is to test if the programmatic policy has learned how to perform goal navigation. Since a new target position can be quite different than the three goals in Ant Cross Maze, we remove all the walls from the environment so the Ant can navigate freely. Fig.20a and 20b depicts the *UP* and

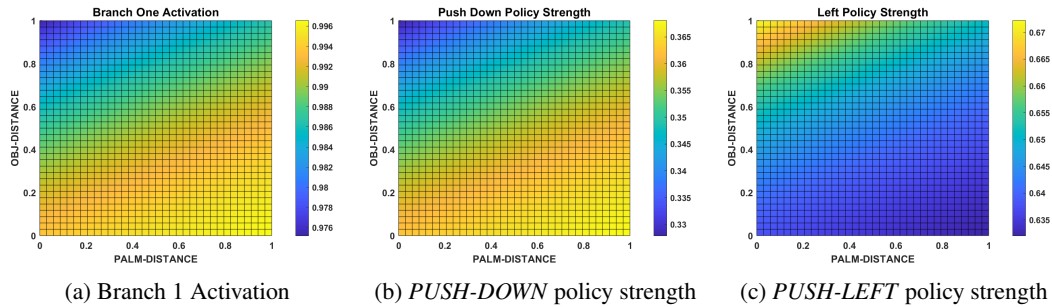

(a) Branch 1 Activation     (b) *PUSH-DOWN* policy strength     (c) *PUSH-LEFT* policy strength

Figure 18: Branch activation and policy strength graph of a pusher program $\mathcal{P}_{pusher}$.

$$95\% \cdot \pi_{\text{FORWARD}}(s) + 5\% \cdot \pi_{\text{JUMP}}(s)$$

$$\mathcal{X} = [\ \mathcal{H}_{next},\ |x_{back} - \mathcal{H}_{next}|\ ]$$

Figure 19: A HalfCheetah Hurdle program $\mathcal{P}_{hc}$ powered by two primitive skill policies: $\pi_{\text{FORWARD}}(s)$ and $\pi_{\text{JUMP}}(s)$. Program input $\mathcal{X}$ includes the position of the next hurdle $\mathcal{H}_{next}$ and the distance from HalfCheetah's back foot to next hurdle $|x_{back} - \mathcal{H}_{next}|$. The program $\mathcal{P}_{hc}$ shown is a special program which does not contain any **if-then-else** structure, i.e., there exists a fixed strength of invoking each primitive policy.

*RIGHT* policy strength if we set the goal to (14, -8), i.e., the upper-right corner. Fig. 20c and 20d shows the *UP* and *LEFT* policy strength if we set the goal to (14, 8), i.e., the upper-left corner. $\mathcal{P}_{cross}$ can successfully drive the Ant to these positions. From these policy strength figures we can conclude that $\mathcal{P}_{cross}$ successfully detects the change of goals and weighs *UP* and associated *RIGHT* or *LEFT* policies in appropriate areas to navigate the ant to reach the goals.

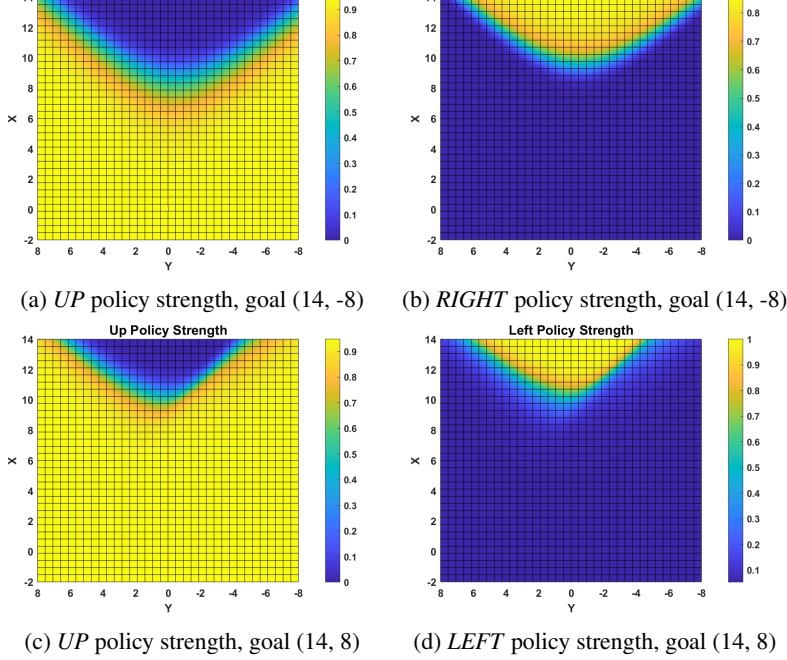

(a) *UP* policy strength, goal (14, -8)     (b) *RIGHT* policy strength, goal (14, -8)

(c) *UP* policy strength, goal (14, 8)     (d) *LEFT* policy strength, goal (14, 8)

Figure 20: *UP*, *RIGHT* and *LEFT* policy strength of a transferred program $\mathcal{P}_{cross}$.

| Goal Radius | $\mathcal{P}_{cross}$ | Comp-SAC |
|:-----------:|:--------------------:|:--------:|
| 1.0 | **0.16±0.04** | 0.28±0.08 |
| 1.5 | **0.13±0.03** | 0.26±0.06 |
| 2.0 | **0.10±0.03** | 0.23±0.05 |

Table 8: Comparison results of policy generalizability in Ant Reshaped Maze environment (Fig. 21). We set multiple goal radius 1.0 (the original radius in Ant Cross Maze), 1.5, and 2.0. Normalized final distances to goals are reported as the average over 50 random executions.

## J.2 ANT RESHAPED MAZE

Furthermore, we consider a reshaped cross maze with three different goals: (8, 8), (14, 0), (8, -8) (represented by blue spheres in Fig.21). The shapes and positions of walls have also been changed and adapted to new possible goals accordingly. This novel environment Ant Reshaped Maze is similar to Ant Cross Maze environment in many ways, e.g. they both have three possible goals, their goals are symmetric, etc.

Figure 21: Ant Reshaped Maze

We directly apply $\mathcal{P}_{cross}$ to Ant Reshaped Maze and compare the generalizability of $\mathcal{P}_{cross}$ with the BiLSTM policy learned in the Ant Cross Maze environment as well using Composition-SAC (Qureshi et al., 2020). We compare with Composition-SAC because the two policies both learn to compose primtivie skills to adapt to new complex behavior. However, $\mathcal{P}_{cross}$ is more interpretable than a BiLSTM model. Table 8 shows that $\mathcal{P}_{cross}$ successfully realizes the change of the three goals and drive the Ant in the novel environment with 0.16 normalized distance to the new goals. In constrast, the Composition-SAC policy does not solve the novel environment well. We further increase the original radius of goals to 1.5 and 2 in the novel environment, both $\mathcal{P}_{cross}$ and the Composition-SAC policy get close to goals with $\mathcal{P}_{cross}$ continuing to perform much better.

## K AFFINE POLICIES AND PID POLICIES

### K.1 AFFINE POLICIES

The DSL for affine policies implements a controller $C$ as affine transformations in Fig. 1:

$$C_{\textit{Affine}} ::= \theta_c + \theta \cdot \mathcal{X} \mid \theta_c$$

where $\theta \in \mathbb{R}^{m \cdot |\mathcal{X}|}, \theta_c \in \mathbb{R}^m$ are control parameters. Particularly, $C_{\textit{Affine}}$ can be as simple as some (learned) constants $\theta_c$.

For example, consider the continuous OpenAI Gym MountainCar environment in Fig. 22a. A car is on a one-dimensional track, positioned between two "mountains". The goal is to drive up the mountain on the right. Because the car's engine is not strong enough, a policy has to drive the car back and forth to build up momentum in multiple passes. The reward is greater if the policy spends less energy to reach the goal.

A learned programmatic policy to control the continuous MountainCar environment is given in Fig. 23. Fig. 22b depicts the activation of branch 1 in the program as a function of $(position, \ velocity)$. The degree of activation (yellow) is close to 1 on almost all states under with speed velocity greater than 0. The logic of the program is then obvious — when ever the car speed is greater than 0, the policy accelerates the forth to drive the car forward; when ever the car speed is less than 0, the policy accelerates the forth to drive the car backward. Indeed, this helps drive the car back and forth to build up momentum. The policy succeeds reaching the goal and receives cumulative reward greater than the threshold 90 on all experienced episodes.

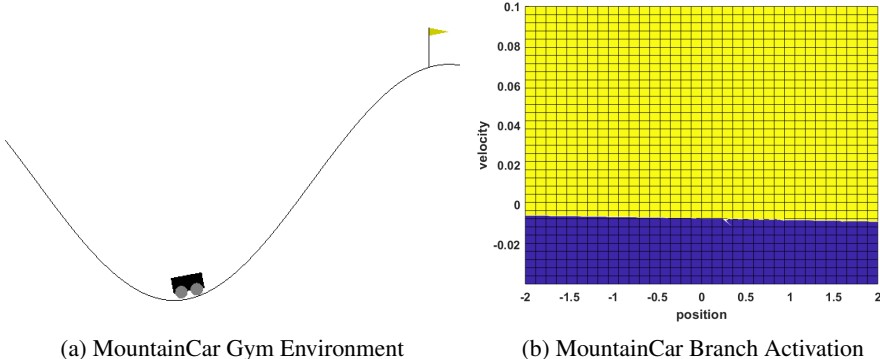

(a) MountainCar Gym Environment        (b) MountainCar Branch Activation

Figure 22: Continuous MountainCar Example. In the branch activation graph, a yellow region indicates the area of states on which the control signal in branch 1 is activated.

---

$$\textbf{if } \theta_{1_c} + \theta_1^T \cdot \mathcal{X} > 0$$
$$\textbf{then } 0.92 \longleftarrow \textbf{Branch 1}$$
$$\textbf{else } -0.95 \longleftarrow \textbf{Branch 2}$$

$$\mathcal{X} = [\, position, \ velocity \,]$$
$$\theta_1 = [\, 0.1, \ 128.1 \,], \ \theta_{1_c} = 0.88$$

Figure 23: A MountainCar program with two branches. Program input $\mathcal{X}$ includes current car *position* and *velocity*.

---

## K.2 PID POLICIES

The DSL for PID policies implements a controller $C$ as a discretized, multivariable PID function in Fig. 1:

$$C_{PID} ::= \textbf{PID}_{\theta_P,\theta_I,\theta_D}(\epsilon, h, s) \mid \theta_c$$

where $\theta_P$, $\theta_I$, $\theta_D \in \mathbb{R}^{m \cdot n}$ are parameters representing the proportional gain, integral gain, and derivative gain matrices of PID control. Notice that a PID controller additionally takes a known constant $\epsilon$ that represents a fixed target for the controller to stabilize the system under control around, and a histroy $h$ of a sequence of states before the current control step. The semantics of the controller is as follows:

$$[\![\textbf{PID}_{\theta_P,\theta_I,\theta_D}]\!](\epsilon, h, s) = \theta_P \cdot P + \theta_I \cdot I + \theta_D \cdot D \text{ where}$$
$$P = (\epsilon - s) \qquad I = \textbf{fold}(+, \epsilon - h) \qquad D = \textbf{peek}(h, -1) - s \tag{13}$$

In the semantics definition, $P$ is the proportional term, $I$ is the discrete approximation of the integral term (calculated via a **fold**), and $D$ is the finite-difference approximation of the derivative term. In line with the standard integral error reset strategy (Åström & Hägglund, 1984), the **fold** function acts over a fixed-sized window on the history (e.g. the five latest states of the history). **peek**$(h, -1)$ returns the most resent state in a history $h$.

For example, consider the OpenAI Gym Pendulum environment in Fig. 24a. The pendulum swingup problem is a classic problem in the control literature. The pendulum starts in a random position, and the goal is to swing it up so it stays upright. The state space include $\omega$ that is the angle the pendulum makes with the vertical and the angular velocity $\dot{\omega}$. However, the observation exposed to a policy for this task is $[cos(\omega), \ sin(\omega), \dot{\omega}]$. Since the goal is to stablise the pendulum upright, the fixed goal $\epsilon$ for a PID controller for this task is $[1, 0, 0]$ (i.e. $[cos(\omega) = 1, \ sin(\omega) = 0, \dot{\omega} = 0]$).

A learned programmatic policy to control the pendulum environment is given in Fig. 25. Fig. 24b depicts the activation of branch 1 in the program as a function of $(\omega, \dot{\omega})$. The degree of activation (yellow) is close to 1 on all states where the PID controller in branch 1 would be applied to. Accordingly, the large constant control signal in branch 2 would be applied states within the blue

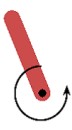

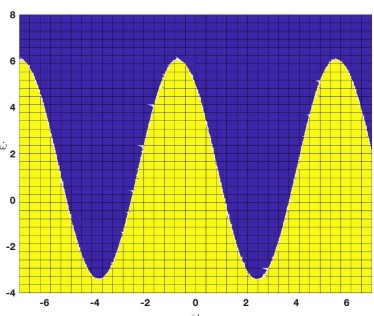

(a) Pendulum Gym Environment          (b) Pendulum Branch Activation

Figure 24: Pendulum Example. In the branch activation graph, a yellow region indicates the area of states on which the control signal in branch 1 is activated.

---

**if** $\theta_{1_c} + \theta_1^T \cdot \mathcal{X} > 0$
    **then** $\text{PID}_{\theta_P, \theta_I, \theta_D}\left( [1, 0, 0], h, s \right) \longleftarrow$ **Branch 1**
    **else** $2.39 \longleftarrow$ **Branch 2**

$\mathcal{X} = [\, cos(\omega),\ sin(\omega),\ \dot{\omega}\, ]$
$\theta_1 = [\, 2.57,\ -2.18,\ -0.71\, ],\ \theta_{1_c} = 0.94$
$\theta_P = [\, -9.28, 1.04, 1.71\, ]^T\ \theta_I = [\, -0.26, 1.57, 0.14\, ]^T\ \theta_D = [\, -4.70, 2.03, -0.48\, ]^T$

Figure 25: A pendulum program with two branches. In the system, $\omega$ is the angle the pendulum makes with the vertical. Program input $\mathcal{X}$ includes $cos(\omega),\ sin(\omega), \dot{\omega}$.

---

region. The logic of the program is then obvious — when ever the pendulum is hanging or downwards close to $+\pi$ or $-\pi$, the policy applies the large (constant) control action in branch 2 to push the pendulum to an area in which it is easy for the PID controller in branch 1 to eventually stabilise the pendulum upright at $0, 2\pi, or -2\pi$. The policy succeeds on all experienced episodes. In our experience, we found that it is impossible to train a single PID controller to solve the task. The programmatic policy in turn uses a large control action to push the pendulum to where the PID controller is able to function.

**Experiment Results of Learning Affine and PID Policies.** We have applied the learning algorithm in Sec. 3 to the popular benchmark suite MuJoCo. This benchmark suite consists of multiple locomotion tasks with 2D and 3D agents. Table 9 demonstrates that our results on Humanoid-v3 and LunarLander-v2 are competitive with state-of-the-art RL algorithms (trained using 3 million environment steps). More results were given in Table 3.

| Environment | $\pi$-RPL | | SAC | | PPO | |
|---|---|---|---|---|---|---|
| | stoc | mean | stoc | mean | stoc | mean |
| Humanoid-v3 | 5207 | 5737 | 5488 | 5521 | 1340 | 1912 |
| LunarLander-v2* | 271 | 279 | 279 | 279 | 16 | 102 |

Table 9: We compare the final episode reward of our programmatic policies $\pi$-RPL with that of SAC (Haarnoja et al., 2018) and PPO (Schulman et al., 2017) agents on Humanoid-v3 and LunarLander-v2. We report the averaged results of three repeated experiments where "stoc" refers to a Gaussian policy with actions sampled from it, while "mean" refers to using mean of the policy. LunarLander(continuous) is controlled by a discretized, multivariable PID program.

