# OpenReview forum: "Programmatic Reinforcement Learning without Oracles"
_ICLR.cc/2022/Conference — ICLR 2022 Spotlight_

### Official Review · Reviewer_xiEe · 2021-10-29

**Correctness:** 3
**Technical Novelty And Significance:** 3
**Empirical Novelty And Significance:** 3
**Recommendation:** 8
**Confidence:** 5

**Main Review:**

## **Paper Strengths**

- **Clarity:** The paper is pretty clear wrt the high-level topic and what the high-level intuitions are. This is good, because I think some low-level parts of the paper could be clearer (see weaknesses section). Figure 3 is also a great and clear figure, and therefore should be referenced in the "Ensemble Policies" subsection of section 2.
- **Experiments:** Comparing against the HRL style of environments such as the Ant mazes and HalfCheetah hurdle is nice, and serves as a nice comparison (although not direct because the abstractions provided are very different) to existing deep RL HRL algorithms. Furthermore, the authors evaluate over 10 random seeds, and include learning curves (not always given for these types of program RL papers), which is very thorough.
- **Method**: This continuous relaxation of the discrete program search tree is neat and intuitive, and serves as a contribution in the program synthesis for RL space.
- **Motivation**: The motivation is nice, to generate interpretable/stable policies for RL by synthesizing programs which can act as the agent in environments. Part of the novelty here is the continuous relaxation, which makes the program search more feasible.
- **Comprehensive Appendix:** The appendix has some solid extra analysis along with example programs for each of the tasks.

## **Paper Weaknesses**

- **Clarity:** I think section 2 can be made a little clearer to people not familiar with the program synthesis domain by introducing examples earlier (e.g. giving a clear example immediately after "production rules X → \sigma_1, \sigma_2,....or nonterminals"). $\phi_{Act}$ could be better defined by first defining $dx_i$, $dy_i$, and using underbraces with text or having an extra sentence explaining it. It took me more time than it should've to parse. Furthermore, I think there should be an algorithm pseudocode box where all stages of training the method for the different versions of the method are detailed, this would make it clearer how exactly the method works (it can be hard to keep track of high-level details when reading a long method section that goes over all individual parts). Finally, your graphs don't have axis labels (Figure 8).
- **Method:** Seems to me that the tree structure is essentially decided beforehand. Essentially one would have to decide how complex the program is a priori, and then learn via policy optimization what the weights of that program tree are. This seems somewhat limiting to me, but in one sense perhaps acts as a hyperparameter that regularizes the policy structure. Could the authors do an ablation study on this with extra analysis?
- **Abstractions:** Having abstractions for Program Synthesis is perfectly reasonable, and even more reasonable in the case of this HRL-style of RL program synthesis. But is $\phi_{\text{Guard}}$ necessary? It seems like a pretty restrictive assumption to me, and would be interesting to see how well the model-based HRL version of the algorithm can work without this guard.
- **Misc:** "The high interpretability of π-PRL policies in turn leads to strong generalizability to novel environments." → The connection between interpretability and generalizability in novel environments is not well explained. Do the authors mean the inductive biases of the program structure enforced by the fact that these have to be simple, interpretable policies?
- **Missing related work:** This is not that big of a deal as both of these papers are recent on arXiv (2021). But authors should cite [1] and [2], and compare/contrast with them in the related works and intro section. But I believe that both of these don't require RL oracles, so perhaps a bit more nuance is needed in the intro/abstract/related work describing the contribution of this paper.
- **Experiments:** In light of the above complaint, I think that generally the experiments lack solid analysis in the main paper. Especially in the high level planning experiments, the authors should explain why they believe HIRO gets stuck where \pi-HPRL does not, why \pi-HPRL is more sample efficient, etc. Furthermore, I think an ablation study on tree depth would be illuminating (mentioned in method weaknesses).
- **Baselines**: Can the authors compare to a program RL baseline? Some of the cited works in the related works should work fine, for example the cited Verma works, the cited Bastani/Silver decision tree papers, or [2]? Even if the specific DSL here won't work with them, or they require teacher policies, this type of comparison would elucidate where some of \pi-PRL/HPRL's advantages come from (the DSL? the program structure? the search procedure? the continuous relaxation? the lack of needing a teacher oracle??).

## **Questions**

- Being not familiar with Composition-SAC, why Composition-SAC instead of just regular SAC?
- "The $\pi^\Tau-$PRL models exhibit less data-efficiency than the Composition-SAC policies in part due to the use of TRPO." Why not use off-policy RL for the method then?

## Minor Issues

- Page 2: "defined for each DSL constructs" → each DSL construct
- Page 4: "u_e contain more than one architecture" → u_e contains
- Page 7: "primitives e.g. Ant turning around at a corner. We" → "primitives, e.g. Ant turning around a corner, we learn a ..."
- Page 7: "Expect pusher" → "Except pusher"
- Although our DSLs are simple, the expressiveness is equivalent
to previous interpretable RL works (Bastani et al., 2018; Verma et al., 2018; 2019) → in light of [1] this sentence may need to be reworded

[1] Dweep Trivedi, Jesse Zhang, Shao-Hua Sun, and Joseph J. Lim. Learning to Synthesize Programs as Interpretable and Generalizable Policies. 2021.

[2] Yichen Yang, Jeevana Priya Inala, Osbert Bastani, Yewen Pu, Armando Solar-Lezama, Martin Rinard. Program Synthesis Guided Reinforcement Learning. 2021.



**Summary Of The Paper:**

This paper presents a method for synthesizing programs to solve MDPs without the need for teacher oracle policies by proposing a continuous program search relaxation that allows synthesizing performant programs as policies.


**Summary Of The Review:**

In summary, I like the high level idea and motivations of this paper. The method is also seemingly novel and performs well. But there are some issues with detail clarity, experiments and analysis, and a comparison to similar baselines. I am currently voting for rejection, but will happily change my score and/or be responsive to authors during the review process as these concerns are addressed.


UPDATE: Score has been revised after discussion.

---

> ### Author Response · Authors · 2021-11-16
> **Response to Reviewer xiEe (1/3)**
>
> We appreciate your insightful feedback and constructive comments! We present our response to each of your concerns and questions below.
>
> **1. Comparison with oracle-guided programmatic RL baselines**
>
> One of the main contributions of our paper is direct programmatic policy search. Existing oracle-guided programmatic RL algorithms suffer from a nontrivial distillation gap as the distillation process can yield suboptimal programmatic policies whose reward performance is significantly worse than that of their oracles. This is because oracles (e.g. neural policies) and programs may reside in very different policy structure spaces. Due to the structural constraints, the program that best imitates an oracle is not necessarily a performant programmatic policy, or could even be much worse than the optimal program in the search space. To address this limitation, we propose programmatic RL solely based on reward signals without using any oracles. This allows the synthesis algorithm to search for optimal programs in the differentiable programmatic policy space, without hindrance. We followed your suggestion to compare our programmatic RL algorithm with oracle-based programmatic RL algorithms that learn policies in the form of programs (using the same DSL as ours) [1] and decision trees [3]. We first report results on Mujoco/OpenAI environments where programmatic policies invoke a set of affine controllers under different environmental conditions. To ensure a fair comparison, we relax the semantics of oracle-guided RL programs to be continuous as well but decision trees remain discrete. Neural oracles and our programmatic policies were trained with TRPO 3 million steps and we report the averaged final reward performance of three repeated experiments.
>
> | Environments | Neural Oracles | Oracle-guided RL Programs | Decision Trees | Oracle-free RL Programs |
> | ----------- | ----------- | ----------- | ----------- | ----------- |
> | Reacher      | -3.93| -8.11| -4.79| -5.81|
> | Walker2d     | 5525 | 3480 | 1330 | 5536 |
> | Hopper       | 3619 | 577  | 2053 | 3220 |
> | HalfCheetah  | 4176 | 3245 | 2810 | 3929 |
> | Ant          | 4675 | 3766 | 3504 | 5568 |
> | Swimmer      | 121  | 122  | 124  | 363  |
> | BipedalWalker| 262  | 260  | 252  | 298  |
> | Pendulum     | -150 | -187 | -171 | -151 |
>
> The reward performance of our oracle-free programmatic RL policies is comparable to, sometimes better than neural network policies. Due to the distillation gap, the oracle-guided RL programs learned by the baselines perform worse than their oracles and our policies. We also tried the imitation-projected programmatic RL algorithm in [2] that optimizes a programmatic policy by taking a mirror gradient descent in a policy space with a mix of neural and programmatic representations, which enables deep policy gradient approaches on programmatic policies. However, our own implementation of the algorithm does not achieve results better than [1] - optimization in the mixed policy space does not lead to significant policy improvement.
>
> We further compared our algorithm with [1] on the environments reported in Sec 5. According to our results in Fig. 8a, these environments are best solved using policies that invoke a set of ensemble controllers. We report the mean normalized final distances to goals of learned policies with standard deviations. The neural oracles were trained by [Compositional SAC](https://arxiv.org/abs/1905.10681).
>
> | Environments | Neural Oracles | Oracle-guided RL Programs | Oracle-free RL Programs |
> | ----------- | ----------- | ----------- | ----------- |
> | Ant Cross Maze     | 0.11$\pm$0.05 | 0.38$\pm$0.15 | 0.10$\pm$0.06 |
> | Ant Random Goal    | 0.12$\pm$0.02 | 0.44$\pm$0.22 | 0.14$\pm$0.04 |
> | HalfCheetah Hurdle | 0.43$\pm$0.22 | 0.47$\pm$0.17 | 0.03$\pm$0.01 |
> | Pusher             | 0.16$\pm$0.06 | 0.24$\pm$0.13 | 0.09$\pm$0.05 |
>
> The oracle-guided RL programs are again suboptimal due to the distillation gap. Especially, they do not work well in environments with multiple goals. For example, we found that although on Ant Cross Maze the neural compositional-SAC policy in general works well, the success rate of reaching the goal (6, -6) is higher than reaching the other goals. The bias is amplified in the oracle-guided programmatic RL policy. When the goal is at (12, 0), the agent sometimes mistakenly goes somewhere near (6, -6).
>
> The above two sets of experiments demonstrate that our oracle-free programmatic RL overcomes the suboptimality induced by oracle-guided programmatic RL.
>
> [1] Abhinav Verma, Vijayaraghavan Murali, Rishabh Singh, Pushmeet Kohli, and Swarat Chaudhuri. Programmatically interpretable reinforcement learning. ICML 2018.
>
> [2] Abhinav Verma, Hoang Minh Le, Yisong Yue, and Swarat Chaudhuri. Imitation-projected programmatic reinforcement learning. NeurIPS 2019.
>
> [3] Osbert Bastani, Yewen Pu, and Armando Solar-Lezama. Verifiable reinforcement learning via policy extraction. NeurIPS 2018.

---

> > ### Author Response · Authors · 2021-11-16
> > **Response to Reviewer xiEe (2/3)**
> >
> > **2. Ablation study on tree depth**
> >
> > The depth bound $d$ on a program derivation tree is indeed a hyperparameter of our policy search algorithm. We encode program architecture search as learning the probability distribution over all programs with their abstract syntax tree depth lower than this bound. We followed your suggestion to conduct an ablation study on the tree depth bound.
> >
> > | | max depth 5 | | max depth 6 | | max depth 7 | |
> > | ----------- | ----------- | ----------- | ----------- | ----------- | ----------- | ----------- |
> > | Environments     | performance | depth | performance | depth | performance | depth|
> > |  Ant Cross Maze   | 0.10$\pm$0.06 | 4.2 | 0.10$\pm$0.06 | 4.75 | 0.07$\pm$0.05 | 5.6 |
> > |  Ant Random Goal    | 0.16$\pm$0.03 | 4.4 | 0.14$\pm$0.04 | 5.3 | 0.15$\pm$0.05 | 5.2 |
> > |  HalfCheetah Hurdle    | 0.03$\pm$0.01 | 3.1 | 0.03$\pm$0.01 | 3.1 | 0.04$\pm$0.02 | 2.6 |
> > |  Pusher   | 0.08$\pm$0.04 | 3.0 | 0.09$\pm$0.05 | 3.5 | 0.10$\pm$0.04 | 4.0 |
> >
> > The performance section compares the mean normalized final distances to goals of learned policies with standard deviations (averaged over 10 random seeds). The result demonstrated that the final performance of the synthesized policies is not sensitive to the bound hyperparameter $d$. Although a larger value of $d$ incurs a greater search space, the algorithm eventually converges to similar architectures. This suggests that if the user does not know how complex the programmatic policy should be a priori, the user can simply set a large tree-depth bound and our algorithm would assign the highest probability to the architecture within the program derivation tree that maximizes cumulative RL reward.
> >
> >
> >
> > **3. The role of the abstraction $\varphi_\text{Guard}$:**
> >
> > We appreciate that you find the abstractions we introduced into the HRL-style program synthesis reasonable. $\varphi_\text{Guard}$ encodes conditions under which an abstract action (e.g. ant moving up or down) can occur. If we do not encode $\varphi_\text{Guard}$ in the constraint for high-level planning, a constraint satisfiability solver may synthesize a high-level plan in which an abstract action cannot be realized in the real environment. For example, in Ant $\supset$-Maze, if $\varphi_\text{Guard}$ is ignored, the solver could simply guide the Ant to move up to the goal directly even there are walls in between. We note that the recent work [2] also requires a logical predicate to be provided for each abstract function in a DSL that encodes the intended behavior of the abstract function.
> >
> > We also appreciate your insightful suggestion of pursuing how well our model-based HRL algorithm can work without $\varphi_\text{Guard}$. Although we cannot directly apply this idea (due to the reason above), it is a very interesting research direction! To this end, we could equip our algorithm with an outer-level counterexample-guided abstraction refinement loop. Initially,  $\varphi_\text{Guard} = \text{True}$ i.e. no guard constraint is enforced. At each iteration, if the algorithm cannot learn a program to realize the high-level plan $p$ synthesized by a constraint solver, we refine $\varphi_\text{Guard} = \varphi_\text{Guard} \wedge \neg gen(p)$ where $gen(p)$ extracts the root cause of why $p$ cannot be implemented e.g. at which abstract state the agent gets stuck and hence the state should be unreachable. The algorithm converges to the optimal high-level plan by iteratively synthesizing new plans and automatically learning $\varphi_\text{Guard}$ on the fly.
> >
> >
> >
> > **4. The relation between interpretability and generalizability**
> >
> > A programmatic policy is more likely to generalize to unseen environments because of the inductive bias implicitly encoded in the policy's interpretable representation, e.g. interpretable if-then-else conditions used to drive Ant to different regions on a maze. We provided a concrete example in Appendix E.2 where we show the programmatic policy learned on Ant Cross Maze generalizes to a maze with a reshaped layout on which the neural compositional-SAC agent fails to generalize.

---

> > > ### Author Response · Authors · 2021-11-16
> > > **Response to Reviewer xiEe (3/3)**
> > >
> > > **5. Missing related work [1] and [2]**
> > >
> > > We will update the paper to discuss [1] and [2]. Briefly, [1] first learns a program embedding space (that supports smooth interpolation and where nearby latent programs correspond to similar execution behaviors) and then searches over the learned program embedding space to synthesize a program for a given task. Our approach differs in the sense that we do not need to prepare a dataset of programs to train the program embedding space. Our tool is more suitable when sampling such a dataset of unique programs is challenging (e.g. Ant continuous control). [2] uses a MaxSAT solver to synthesize straight-line programs to reach a goal. Our method is more suitable when there are multiple goals in an environment for which program architecture search is necessary to synthesize conditional branches to reach diverse goals.
> > >
> > > [1] Dweep Trivedi, Jesse Zhang, Shao-Hua Sun, and Joseph J. Lim. Learning to Synthesize Programs as Interpretable and Generalizable Policies. 2021.
> > >
> > > [2] Yichen Yang, Jeevana Priya Inala, Osbert Bastani, Yewen Pu, Armando Solar-Lezama, Martin Rinard. Program Synthesis Guided Reinforcement Learning. 2021.
> > >
> > >
> > >
> > > **6. Comparison between $\pi$-HPRL and HIRO**
> > >
> > > In our experiments, the HIRO agents get stuck in a local optimum as they do not reach the goals. Unlike $\pi$-HPRL, HIRO is model-free and does not know the structure of the (abstract) state space, so it is unable to discover the path from the initial region to the goal region. $\pi$-HPRL is more sample-efficient because it uses the available (abstract) structure of an environment and the abstract specifications of primitives to synthesize a high-level control plan. Such a control plan consists of a sequence of sub tasks (e.g. Fig. 6c for Ant $\supset$-Maze) and each sub task is in accordance with the primitive specifications e.g. Ant moving up or turning around at a corner to a sub goal. A sub task is easy to train because (1) a sub goal is easy to achieve by utilizing the corresponding primitives and (2) the training procedure is guided by dense distance-based rewards to the sub goal. Therefore, training for each sub task converges extremely fast well in advance of the expiration of the training budget allocated to the sub task, causing the ladder-shape convergence curves in Fig. 8b.
> > >
> > >
> > >
> > > **7. Why use Compositional SAC instead of regular SAC as a baseline?**
> > >
> > > We compared our tool with Compositional SAC because it also learns to compose multiple task-agnostic primitives to solve task environments but instead uses a Bidirectional LSTM-based agent. We aim to demonstrate that a programmatic agent is more interpretable and generalizable than an LSTM agent. Additionally, compositional SAC outperforms SAC on the benchmarks in the paper.
> > >
> > >
> > >
> > > **8. Why not use off-policy RL for the method?**
> > >
> > > We use TRPO to minimize the surrogate objectives in Equations 2 and 3. Our approach can be extended to off-policy RL e.g. alternatively updating the policy architecture weights and policy parameters of a program derivation tree to minimize the KL-divergence constraint in SAC. We leave the extension as future work.
> > >
> > >
> > >
> > > **9. Clarity**
> > >
> > > We thank the reviewer for the helpful suggestions on how to improve the clarity of our paper. We have been revising the paper to incorporate all of them and your other valuable comments. We will submit the updated paper shortly.

---

> > > > ### Comment · Reviewer_xiEe · 2021-11-16
> > > > **Response to Authors' Initial comments**
> > > >
> > > > Thanks for the detailed response. The authors have essentially addressed all my complaints, other than explicitly revising the paper to account for them.
> > > >
> > > > I will patiently await the authors' updated revision, and then adjust my score accordingly based on the actual changes in the revised paper.

---

> > > > > ### Author Response · Authors · 2021-11-19
> > > > > **Paper has been revised.**
> > > > >
> > > > > We would like to express our deepest gratitude for your patience! We have revised our manuscript in response to your questions and concerns (all changed parts are colored in blue for visibility):
> > > > >
> > > > > * We have added a comparison with oracle-guided programmatic RL baselines in Sec 5 (Table 2 and the second paragraph of page 9) and Appendix C. The result confirms that our oracle-free programmatic RL overcomes the suboptimality induced by oracle-guided programmatic RL.
> > > > > * We have added an ablation study on tree depths in Sec 5 (Table 2 and the first paragraph of page 9) and Appendix B. Parciturlarly, we have included the convergence curves of this ablation study in Fig. 9 of the appendix. We found that the final performance of the synthesized policies is not sensitive to the depth-bound hyperparameter.
> > > > > * We have clarified the relation between interpretability and generalizability at the bottom of page 7.
> > > > > * We have added a discussion about the missing related work in the first paragraph of Sec 1 and Sec 6 (page 9).
> > > > > * We have added a discussion about why HIRO gets stuck where $\pi$-HPRL does not and why $\pi$-HPRL is more sample efficient at the bottom of page 8.
> > > > > * We have improved the clarify our paper by (a) giving an example immediately after "production rules $X::= \sigma_1, \sigma_2,....$ or nonterminals" in Sec 2, (b) redefining and explaining $\varphi_\text{Act}$ in Sec 4, (c) adding the pseudocode for both the programmatic RL architecture search algorithm (referred to in Sec 3) and programmatic high-level planning algorithm (referred to in Sec 4) in Appendix A, and (d) adding axis labels to all figures in the paper.
> > > > >
> > > > > In conclusion, we have performed new experiments and revised our manuscript following your valuable suggestions, which we feel significantly strengthen the paper. For this, we greatly appreciate your insightful comments. Please let us know if the revised paper has adequately clarified your concerns.

---

> > > > > > ### Comment · Reviewer_xiEe · 2021-11-20
> > > > > > **Revised paper summary**
> > > > > >
> > > > > > Thank you for addressing my concerns, I have looked at the revised manuscript and updated my score in response.

---

### Official Review · Reviewer_iSsz · 2021-11-01

**Correctness:** 4
**Technical Novelty And Significance:** 3
**Empirical Novelty And Significance:** 3
**Recommendation:** 8
**Confidence:** 4

**Main Review:**

The idea of using a differentiable version of a DSL to allow for gradient-based optimization is clever and elegant. I understand that the authors were inspired by methods used to learn neural network architectures such as DARTs. The application to the synthesis of programmatic policies is non-trivial and interesting, thus worthy of publication.

However, I have a two concerns with the paper.

The first concern is minor. Why is the title of the paper "Programmatic Reinforcement Learning Without Oracles"? In my opinion, the main contribution of the work and that makes it interesting is the differentiable approximation of the DSL that allows one to perform gradient methods to search over the originally discrete space of programs. Learning without an oracle can also be achieved without the differentiable approximation. For example, if the problem domains were discrete, one could use an enumeration procedure similar to what NDS does without the Bayesian optimization step [1].

Moreover, the need of having an oracle isn't necessarily a major hurdle. If an oracle was available how would the proposed method compare with other methods such as NDS [1] and Propel [2]? I am wondering if the main advantage of the proposed method is that it doesn't require an oracle for training or if it is able to more quickly find an effective programmatic policy in the large space induced by the DSL.

The second concern I have is regarding the computational complexity of equation defining $\[E\](s)$. The equation is recursive and resolving the value of the equation is equivalent to traversing the tree of programs. This computation can be prohibitive depending on the DSL and on the size of the program that needs to be synthesized. That is, we need to enumerate all programs up to some depth for each state evaluated during an episode!

I understand that the complexity of $\[E\](s)$ doesn't involve gathering samples from the environment, but it has a non-negligible computational complexity that isn't even mentioned in the paper (I apologize if I missed a discussion about the complexity of $\[E\](s)$). What happens if the x-axis of the plots is given in running time? Will the method proposed in the paper be much slower than other competing methods?

The following could be better explained in the paper:

1) Why encode the tree program as a Gaussian distribution?
2) rho isn't defined in Equation 2.
3) Explain that the action the tree program returns is given by the evaluation of the root of the tree with the [E](s) equation.

References

[1] Abhinav Verma, Vijayaraghavan Murali, Rishabh Singh, Pushmeet Kohli, and Swarat Chaudhuri. Programmatically interpretable reinforcement learning. Proceedings of the 35th International Conference on Machine Learning, ICML 2018.

[2] Abhinav Verma, Hoang Minh Le, Yisong Yue, and Swarat Chaudhuri. Imitation-projected programmatic reinforcement learning. Advances in Neural Information Processing Systems 32: Annual Conference on Neural Information Processing Systems 2019, NeurIPS 2019


**Summary Of The Paper:**

This paper presents a novel method for synthesizing programmatic policies. The code idea of the method is to define a relaxed and differentiable version of the domain-specific language (DSL) used to encode the programmatic policies.

The program space the DSL induces can be described as a program of the DSL itself. Since the program defines a differentiable space, one can use policy gradient methods to search in the space of programs by assigning higher probabilities to production rules that maximize the expected rewards.

The search for programmatic policies happens in two steps. The first step applies a policy gradient method to optimize the probabilities defining the program space. Then, one can extract the most likely program structure from the search space by greedily choosing the production rules with higher probabilities. The resulting program is further optimized with reinforcement learning.

Empirical results on continuous control problems show the advantages of the method.

**Summary Of The Review:**

The paper presents a novel and interesting approach to synthesizing programmatic policies that uses a differentiable approximation of a domain-specific language for policies. This differentiable DSL allows for the use of policy gradient methods to search over the space of programs. The main weakness of the method is the computational complexity required to compute the action of the policy during training. The method iterates through all programs one can synthesize in the DSL up to a given depth. The computational complexity of the policy isn't even mentioned in the paper and it can be a show-stopper depending on the DSL and on the size of the program that needs to be synthesized.

---

> ### Author Response · Authors · 2021-11-16
> **Response to Reviewer iSsz (1/2)**
>
> We appreciate your insightful feedback and constructive comments! We present our response to each of your concerns and questions below.
>
> **1. Comparison with oracle-guided programmatic RL baselines**
>
> One of the main contributions of our paper is direct programmatic policy search. Existing oracle-guided programmatic RL algorithms suffer from a nontrivial distillation gap as the distillation process can yield suboptimal programmatic policies whose reward performance is significantly worse than that of their oracles. This is because oracles (e.g. neural policies) and programs may reside in very different policy structure spaces. Due to the structural constraints, the program that best imitates an oracle is not necessarily a performant programmatic policy, or could even be much worse than the optimal program in the search space. To address this limitation, we propose programmatic RL solely based on reward signals without using any oracles. This allows the synthesis algorithm to search for optimal programs in the differentiable programmatic policy space, without hindrance. We followed your suggestion to compare our programmatic RL algorithm with oracle-based programmatic RL algorithms that learn policies in the form of programs (using the same DSL as ours) [1] and decision trees [3]. We first report results on Mujoco/OpenAI environments where programmatic policies invoke a set of affine controllers under different environmental conditions. To ensure a fair comparison, we relax the semantics of oracle-guided RL programs to be continuous as well but decision trees remain discrete. Neural oracles and our programmatic policies were trained with TRPO 3 million steps and we report the averaged final reward performance of three repeated experiments.
>
> | Environments | Neural Oracles | Oracle-guided RL Programs | Decision Trees | Oracle-free RL Programs |
> | ----------- | ----------- | ----------- | ----------- | ----------- |
> | Reacher      | -3.93| -8.11| -4.79| -5.81|
> | Walker2d     | 5525 | 3480 | 1330 | 5536 |
> | Hopper       | 3619 | 577  | 2053 | 3220 |
> | HalfCheetah  | 4176 | 3245 | 2810 | 3929 |
> | Ant          | 4675 | 3766 | 3504 | 5568 |
> | Swimmer      | 121  | 122  | 124  | 363  |
> | BipedalWalker| 262  | 260  | 252  | 298  |
> | Pendulum     | -150 | -187 | -171 | -151 |
>
> The reward performance of our oracle-free programmatic RL policies is comparable to, sometimes better than neural network policies. Due to the distillation gap, the oracle-guided RL programs learned by the baselines perform worse than their oracles and our policies. We also tried the imitation-projected programmatic RL algorithm in [2] that optimizes a programmatic policy by taking a mirror gradient descent in a policy space with a mix of neural and programmatic representations, which enables deep policy gradient approaches on programmatic policies. However, our own implementation of the algorithm does not achieve results better than [1] - optimization in the mixed policy space does not lead to significant policy improvement.
>
> We further compared our algorithm with [1] on the environments reported in Sec 5. According to our results in Fig. 8a, these environments are best solved using policies that invoke a set of ensemble controllers. We report the mean normalized final distances to goals of learned policies with standard deviations. The neural oracles were trained by [Compositional SAC](https://arxiv.org/abs/1905.10681).
>
> | Environments | Neural Oracles | Oracle-guided RL Programs | Oracle-free RL Programs |
> | ----------- | ----------- | ----------- | ----------- |
> | Ant Cross Maze     | 0.11$\pm$0.05 | 0.38$\pm$0.15 | 0.10$\pm$0.06 |
> | Ant Random Goal    | 0.12$\pm$0.02 | 0.44$\pm$0.22 | 0.14$\pm$0.04 |
> | HalfCheetah Hurdle | 0.43$\pm$0.22 | 0.47$\pm$0.17 | 0.03$\pm$0.01 |
> | Pusher             | 0.16$\pm$0.06 | 0.24$\pm$0.13 | 0.09$\pm$0.05 |
>
> The oracle-guided RL programs are again suboptimal due to the distillation gap. Especially, they do not work well in environments with multiple goals. For example, we found that although on Ant Cross Maze the neural compositional-SAC policy in general works well, the success rate of reaching the goal (6, -6) is higher than reaching the other goals. The bias is amplified in the oracle-guided programmatic RL policy. When the goal is at (12, 0), the agent sometimes mistakenly goes somewhere near (6, -6).
>
> [1] Abhinav Verma, Vijayaraghavan Murali, Rishabh Singh, Pushmeet Kohli, and Swarat Chaudhuri. Programmatically interpretable reinforcement learning. ICML 2018.
>
> [2] Abhinav Verma, Hoang Minh Le, Yisong Yue, and Swarat Chaudhuri. Imitation-projected programmatic reinforcement learning. NeurIPS 2019.
>
> [3] Osbert Bastani, Yewen Pu, and Armando Solar-Lezama. Verifiable reinforcement learning via policy extraction. NeurIPS 2018.

---

> > ### Author Response · Authors · 2021-11-16
> > **Response to Reviewer iSsz (2/2)**
> >
> > In a nutshell, the above results show that our oracle-free programmatic RL overcomes the suboptimality induced by oracle-guided programmatic RL (an important contribution of our work). Our experiments also demonstrate that the method can more efficiently find an effective programmatic policy in the large space induced by a DSL than enumerative policy search (Table 1). We will update the title of our paper if the reviewer still finds it confusing and if this is allowed in the review process.
> >
> >
> >
> > **2. Computational complexity of $\[\[E\]\](\cdot)$**
> >
> > We thank the reviewer for pointing out the computation overhead of our approach, which is a tradeoff for automatic policy architecture search. Assume that $d$ is the depth of a program derivation tree with its root as the initial nonterminal $E$, $k$ is the number of production rules in a DSL, and $m$ is the maximum number of nonterminals in the body of any DSL rule. The number of DSL operations (e.g. evaluations of ensemble policies or Boolean conditions) invoked by $\[\[E\]\](\cdot)$ is bounded by $O((km)^d)$. We compare the computational complexity of $\[\[E\]\](\cdot)$ (where $d=6$) with that of the deepest single program in $\[\[E\]\](\cdot)$ and that of the Bidirectional LSTM-based model used by the Compositional SAC baseline as follows. We report the average running time of over 10000 random executions of the three models.
> >
> > | Model      | Ant Cross Maze | Ant Random Goal | Pusher | Cheetah Hurdle |
> > | ----------- | ----------- | ----------- | ----------- | ----------- |
> > | Program derivation tree      | 0.0023s | 0.0023s | 0.0014s | 0.0014s |
> > | Programmatic policy (depth 6)| 0.0019s | 0.0019s | 0.0009s | 0.0009s |
> > | Bidirectional LSTM           | 0.0034s | 0.0033s | 0.0022s | 0.0021s |
> >
> > The data shows that a single programmatic policy runs very fast. ~~Our training method is indeed slower than policy learning with a BiLSTM model, but this is mostly because we uniquely perform architecture search.~~
> >
> > **Update (Nov 17): we have just optimized the runtime execution of a program derivation tree. As depicted in Fig. 5, primitive policies are invoked multiple times by various programs embedded in a program derivation tree. They were called more than once in our implementation. However, since the input to any primitive is always the current environment state, we could call each primitive just once and use its result anywhere it is invoked in the tree policy. The above (updated) experiment results show that this simple optimization could enable us to run our algorithm as fast as the Compositional SAC baseline. We thank the reviewer for the guidance! This optimization does not affect the complexity analysis in our response.**
> >
> > *More importantly, we do not need to explicitly enumerate and execute all programs up to some depth for each state evaluated during an episode*. Fig. 5 shows that we allow the vast number of programs in a program derivation tree to share computation (similar to weight sharing in DARTS) - intermediate results computed by a shallower program can be reused by a deeper program.
> >
> > We will update the paper to address the computational complexity issue. For more complex DSLs with a large number $k$ of production rules, we could optimize the architecture search procedure by applying a strategy similar to [Progressive DARTS](https://arxiv.org/abs/1904.12760). This strategy gradually increases tree depth $d$ during search while dropping the search directions leading to lowest-weighted programs at the previous stage from the program derivation tree. Such a progressive procedure would allow our algorithm to deeply explore the architecture space even when $k$ is large. We left it for future work.
> >
> >
> >
> > **3. Why encode the tree program as a Gaussian distribution?**
> >
> > We encode a program derivation tree as a Gaussian policy (where the tree's output is the mean) because we aimed to optimize the tree program with stochastic policy gradients. We will improve this statement in the paper to resolve the confusion.
> >
> > We will also explain that the action the tree program returns is given by the evaluation of the root of the tree with the $E$ equation.
> >
> >
> >
> > **4. $\rho$ isn't defined in Equation 2.**
> >
> > $\rho_\pi$ is the discounted state visitation frequency of a policy $\pi$. We will define $\rho$ in the paper.
> >
> >
> > We thank the reviewer for the valuable comments and suggestions. We have been revising the paper to incorporate all of them. We will submit the updated paper shortly.

---

> ### Author Response · Authors · 2021-11-20
> **Paper has been revised.**
>
> We would like to express our deepest gratitude for your constructive feedback again! We have revised our manuscript in response to your questions and concerns (all changed parts are colored in blue for visibility):
>
> * We have added a comparison with oracle-guided programmatic RL baselines in Sec 5 (Table 2 and the second paragraph of page 9) and Appendix C. The result confirms that our oracle-free programmatic RL overcomes the suboptimality induced by oracle-guided programmatic RL.
> * We have added a complexity analysis of the run-time execution of a program derivation tree policy in Sec 3 (page 5). We have also included some experimental results and how to optimize the run-time cost in Appendix F.1. The results show that our simple optimization strategy could enable us to run our algorithm as fast as the Compositional SAC baseline. We have also explained in Appendix F.1 that due to computation sharing, each time we do not need to explicitly enumerate and execute all programs in a program derivation tree up to some depth.
> * We have explained in Sec 3 (page 5) that the action a program derivation tree program returns is given by the evaluation of the root of the tree with the $\[\[E\]\]$ equation.
> * We have explained in Sec 3 (page 5) that we encode a program derivation tree program as a Gaussian policy in order to calculate stochastic policy gradients.
> * We have defined that $\rho_\pi$ is the discounted state visitation frequency of a policy $\pi$ in Sec 3 (page 5).
>
> In conclusion, we have performed new experiments and revised our manuscript following your valuable suggestions, which we feel significantly strengthen the paper. For this, we greatly appreciate your insightful comments. Please let us know if the revised paper has adequately clarified your concerns.

---

### Official Review · Reviewer_R2Gf · 2021-11-01

**Correctness:** 3
**Technical Novelty And Significance:** 3
**Empirical Novelty And Significance:** 4
**Recommendation:** 8
**Confidence:** 4

**Main Review:**

## Paper strengths and contributions

**Motivation**
- Exploring using programmatic policies structured in more interpretable representations to yield better interpretability compared to deep RL policies is promising.
- The idea of conducting a program search on a differentiable architecture space with policy gradient is novel and convincing. This paper presents an effective way to implement this idea.

**Technical contribution**
- Relaxing the discrete program architecture search space to be continuous allows for optimizing the program architecture and the parameters of program modules using policy gradient methods.
- To the best of my knowledge, leveraging pre-defined/pre-learned primitive program functions (i.e. ensemble policies) to tackle hierarchical RL problems in the field of programmatic RL is novel.
- Using specified state transition behavior of the abstract actions in HRL training seems effective, especially when simple, task-agnostic skills is available

**Clarity**

The overall writing is clear. The authors utilize figures well to illustrate the presented ideas. Figure 5 clearly shows the idea of the Program Derivation Tree and how it creates differentiable architecture spaces.

**Experimental results**
- The presentation of the experimental results is clear and the discussion is sufficient and mostly convincing.
- The experimental results show that the proposed method outperforms or achieves competitive performance compared to RL baselines such as HIRO, TRPO, PPO.

**Reproducibility**

The code is provided, which helps understand the details of the proposed framework.

## Paper weaknesses and questions

**DSLs without Loops**

The paper simply states that the loops are omitted in this work. Yet, describing many desired behaviors involves loops/repetitive behaviors. I would like to hear the authors' opinions on how the proposed framework can deal with such behavior.

**PRL baselines**

The paper only compares with non-programmatic RL methods.
It would be more informative to compare the proposed method against the PRL works (such as "[Verifiable reinforcement learning via policy extraction](https://arxiv.org/abs/1805.08328)", "[Synthesizing Programmatic Policies that Inductively Generalize](https://openreview.net/forum?id=S1l8oANFDH)", etc.), even they require training RL oracles first, which should not be a critical assumption given the tasks considered in this work. With these comparisons, it would be easier to justify how well the proposed method can perform. Or, it would be even greater to show what suboptimality is induced by learning from RL oracles so that it would make this work more convincing.

**Figure 8b: the ladder-shape convergence curves**

Do authors have a good intuition on why the convergence curves of the proposed method presented in Figure 8b look like that?

**Related work: PRL work that does not use RL oracles**

While I am well aware of the fact that arXiv papers do not count as publications, it would be great if the authors can distinguish this paper from a recent work ("[Learning to Synthesize Programs as Interpretable and Generalizable Policies](https://arxiv.org/abs/2108.13643)") that also tackles learning a programmatic policy purely from rewards. (This suggestion does not affect my evaluation of this paper.)

**Related work: program synthesis**

Discussing prior works in program synthesis, whose aim is to synthesize programs to fulfill some specifications, would make the related work section more comprehensive. Some relevant works include:
- RobustFill: Neural Program Learning under Noisy I/O
- DeepCoder: Learning to Write Programs
- Leveraging Grammar and Reinforcement Learning for Neural Program Synthesis
- Execution-Guided Neural Program Synthesis
- Neural Program Synthesis from Diverse Demonstration Videos
- Learning to Infer and Execute 3D Shape Programs
- Latent Programmer: Discrete Latent Codes for Program Synthesis

**The proposed method without compositionality**

The effectiveness of compositionality is not fully justified since no experiment with the proposed method without ensemble policies is presented.

**Decreased interpretability with the transformation**

While the differentiable architecture space is mentioned as a strength of the paper, it requires affine transformation for conditions in if-else statements, which decreases the interpretability of the program in complex environments and makes it difficult to be interpreted by non-expert users.

**Code**

Please include a requirement.txt specifying the versions of all the dependencies. I have encountered many issues when trying to reproduce the results.

## Other metrics

### Relevance and significance
Solid contributions to a relevant problem

### Novelty
Several novel and surprising contributions

### Technical quality
Technically adequate for its area, solid results

### Experimental evaluation
Insufficient or lacking evaluation in 1-2 criteria, but sufficient w.r.t. the other criteria

### Clarity
Very clear, only minor flaws.

**Summary Of The Paper:**

This paper addresses the problem of the low efficiency of program search guided by a pre-trained oracle or on discrete and non-differentiable architecture space. To this end, the paper proposes a framework that performs program architecture search on top of a differentiable relaxation of the architecture space. This allows the program architectures and parameters to be learned via policy-gradient methods without RL oracle. The proposed method also exploits compositionality by allowing an ensemble of primitive functions that perform task-agnostic skills. The experimental results on navigation and manipulation domains show that the proposed method can reliably obtain task-solving programs and outperforms or performs competitively compared to RL baselines including SAC, PPO, TRPO. I believe this work studies an interesting and promising research direction and proposes a convincing framework to tackle this problem with solid technical contributions. Yet, I am mainly concerned with some missing baselines, relevant works, and ablations.

**Summary Of The Review:**

I believe this work studies an interesting and promising research direction and proposes a convincing framework to tackle this problem with solid technical contributions. Yet, I am mainly concerned with some missing baselines, relevant works, and ablations.

---

> ### Author Response · Authors · 2021-11-16
> **Response to Reviewer R2Gf (1/3)**
>
> We appreciate your insightful feedback and constructive comments! We present our response to each of your concerns and questions below.
>
> **1. Comparison with oracle-guided programmatic RL baselines**
>
> One of the main contributions of our paper is direct programmatic policy search. Existing oracle-guided programmatic RL algorithms suffer from a nontrivial distillation gap as the distillation process can yield suboptimal programmatic policies whose reward performance is significantly worse than that of their oracles. This is because oracles (e.g. neural policies) and programs may reside in very different policy structure spaces. Due to the structural constraints, the program that best imitates an oracle is not necessarily a performant programmatic policy, or could even be much worse than the optimal program in the search space. To address this limitation, we propose programmatic RL solely based on reward signals without using any oracles. This allows the synthesis algorithm to search for optimal programs in the differentiable programmatic policy space, without hindrance. We followed your suggestion to compare our programmatic RL algorithm with oracle-based programmatic RL algorithms that learn policies in the form of programs (using the same DSL as ours) [1] and decision trees [3]. We first report results on Mujoco/OpenAI environments where programmatic policies invoke a set of affine controllers under different environmental conditions. To ensure a fair comparison, we relax the semantics of oracle-guided RL programs to be continuous as well but decision trees remain discrete. Neural oracles and our programmatic policies were trained with TRPO 3 million steps and we report the averaged final reward performance of three repeated experiments.
>
> | Environments | Neural Oracles | Oracle-guided RL Programs | Decision Trees | Oracle-free RL Programs |
> | ----------- | ----------- | ----------- | ----------- | ----------- |
> | Reacher      | -3.93| -8.11| -4.79| -5.81|
> | Walker2d     | 5525 | 3480 | 1330 | 5536 |
> | Hopper       | 3619 | 577  | 2053 | 3220 |
> | HalfCheetah  | 4176 | 3245 | 2810 | 3929 |
> | Ant          | 4675 | 3766 | 3504 | 5568 |
> | Swimmer      | 121  | 122  | 124  | 363  |
> | BipedalWalker| 262  | 260  | 252  | 298  |
> | Pendulum     | -150 | -187 | -171 | -151 |
>
> The reward performance of our oracle-free programmatic RL policies is comparable to, sometimes better than neural network policies. Due to the distillation gap, the oracle-guided RL programs learned by the baselines perform worse than their oracles and our policies. We also tried the imitation-projected programmatic RL algorithm in [2] that optimizes a programmatic policy by taking a mirror gradient descent in a policy space with a mix of neural and programmatic representations, which enables deep policy gradient approaches on programmatic policies. However, our own implementation of the algorithm does not achieve results better than [1] - optimization in the mixed policy space does not lead to significant policy improvement.
>
> We further compared our algorithm with [1] on the environments reported in Sec 5. According to our results in Fig. 8a, these environments are best solved using policies that invoke a set of ensemble controllers. We report the mean normalized final distances to goals of learned policies with standard deviations. The neural oracles were trained by [Compositional SAC](https://arxiv.org/abs/1905.10681).
>
> | Environments | Neural Oracles | Oracle-guided RL Programs | Oracle-free RL Programs |
> | ----------- | ----------- | ----------- | ----------- |
> | Ant Cross Maze     | 0.11$\pm$0.05 | 0.38$\pm$0.15 | 0.10$\pm$0.06 |
> | Ant Random Goal    | 0.12$\pm$0.02 | 0.44$\pm$0.22 | 0.14$\pm$0.04 |
> | HalfCheetah Hurdle | 0.43$\pm$0.22 | 0.47$\pm$0.17 | 0.03$\pm$0.01 |
> | Pusher             | 0.16$\pm$0.06 | 0.24$\pm$0.13 | 0.09$\pm$0.05 |
>
> The oracle-guided RL programs are again suboptimal due to the distillation gap. Especially, they do not work well in environments with multiple goals. For example, we found that although on Ant Cross Maze the neural compositional-SAC policy in general works well, the success rate of reaching the goal (6, -6) is higher than reaching the other goals. The bias is amplified in the oracle-guided programmatic RL policy. When the goal is at (12, 0), the agent sometimes mistakenly goes somewhere near (6, -6).
>
> The above two sets of experiments demonstrate that our oracle-free programmatic RL overcomes the suboptimality induced by oracle-guided programmatic RL.
>
> [1] Abhinav Verma, Vijayaraghavan Murali, Rishabh Singh, Pushmeet Kohli, and Swarat Chaudhuri. Programmatically interpretable reinforcement learning. ICML 2018.
>
> [2] Abhinav Verma, Hoang Minh Le, Yisong Yue, and Swarat Chaudhuri. Imitation-projected programmatic reinforcement learning. NeurIPS 2019.
>
> [3] Osbert Bastani, Yewen Pu, and Armando Solar-Lezama. Verifiable reinforcement learning via policy extraction. NeurIPS 2018.

---

> > ### Author Response · Authors · 2021-11-16
> > **Response to Reviewer R2Gf (2/3)**
> >
> > **2. DSLs without Loops**
> >
> > Our loop-free programmatic policies can already produce repetitive behaviors when necessary. This is because policies are always executed in a feedback loop with RL environments. At each iteration, environment states encountered by repetitive behaviors would repeatedly activate or be matched to the same if-then-else conditions and therefore trigger alike actions. To support this argument, we evaluated our tool on two environments from [Synthesizing Programmatic Policies that Inductively Generalize](https://trustml.github.io/docs/iclr20a.pdf): Car and QuadPO. Both environments need policies that capture repeating behaviors. For Car, the goal is to drive a car out of a parking spot to an adjacent lane while avoiding collisions. For QuadPO, the goal is to maneuver a 2D quadcopter through an obstacle course by controlling its vertical acceleration. The training and test distributions are varied to evaluate whether policies can produce an arbitrary number of repetitions. For example, on QuadPO, the obstacle course length is doubled during testing. We summarize the results on test distributions by measuring the fraction of rollouts (out of 500) that safely reach the goal:
> >
> > | Environments | TRPO+NN | TRPO+Programmatic (affine) Policy |
> > | ----------- | ----------- | ----------- |
> > | Car      | 82.5% | 100% |
> > | QuadPO   | 69.6% | 85.6%|
> >
> > Our (loop-free) programmatic policies generalize better than neural policies on the test distributions. We did not compare our policies with the state-machine policies in their paper as we do not have the implementation.
> >
> > Moreover, our tool indeed supports loops in a program that sequentially processes a history of environment states and actions. For example, our DSL allows a controller $C$ to be expanded as a discretized, multivariable PID controller (Fig. 1). In Appendix F.2, such a PID controller is formalized based on the higher-order combinator *fold* that acts over a fixed-sized window on a history. We could alternatively add *fold(f, h)* directly to the DSL to search policies that combine the results of recursively processing each past state in a history $h$ to build up a return action (the body of the combining operation $f$ can also be synthesized).
> >
> >
> >
> > **3. Figure 8b: the ladder-shape convergence curves**
> >
> > Our approach uses the available (abstract) structure of an environment and the abstract specifications of primitives to synthesize a high-level control plan to discover the path from the initial region to the goal region. Such a control plan consists of a sequence of sub tasks (e.g. Fig. 6c for Ant $\supset$-Maze) and each sub task is in accordance with the primitive specifications e.g. Ant moving up or turning around at a corner to a sub goal. A sub task is easy to train because (1) a sub goal is easy to achieve by utilizing the corresponding primitives and (2) the training procedure is guided by dense distance-based rewards to the sub goal. Therefore, training for each sub task converges extremely fast well in advance of the expiration of the training budget allocated to the sub task, causing the ladder-shape convergence curves in Fig. 8b.
> >
> >
> >
> > **4. Decreased interpretability in complex environments**
> >
> > The DSL grammars in the paper use affine transformation for conditions in if-else expressions. However, our algorithm is not specific to this DSL. Instead, it can take any policy DSL with differentiable semantics as input. To increase interpretability, the user could provide a DSL in which if-else conditions are restricted to the form $c_1 \cdot x_i \ge c2$ where $x_i$ is a state variable and $c_1, c_2$ are (synthesized) constants. In this case, the user may have to increase the maximum depth of the program derivation tree to synthesize optimal policies in a deeper search space (as a tradeoff for greater interpretability).
> >
> >
> >
> > **5. Related work on programmatic RL without oracles**
> >
> > We will update the paper to discuss [1]. Briefly, [1] first learns a program embedding space (that supports smooth interpolation and where nearby latent programs correspond to similar execution behaviors) and then searches over the learned program embedding space to synthesize a program for a given task. Our approach differs in the sense that we do not need to prepare a dataset of programs to train the program embedding space. Our tool is more suitable when sampling such a dataset of unique programs is difficult (e.g. Ant continuous control).
> >
> > [1] Dweep Trivedi, Jesse Zhang, Shao-Hua Sun, and Joseph J. Lim. Learning to Synthesize Programs as Interpretable and Generalizable Policies. 2021.

---

> > > ### Author Response · Authors · 2021-11-16
> > > **Response to Reviewer R2Gf (3/3)**
> > >
> > > **6. Related work on neural program synthesis**
> > >
> > > We appreciate your suggestion to discuss neural program synthesis in the paper. We will edit the related work section to include a discussion. Briefly, our method differs from neural program synthesis in the sense that we do not require a dataset of diverse programs and their inputs/outputs for pretraining, so is more suitable in the RL environments we consider.
> > >
> > >
> > >
> > > **7. Code**
> > >
> > > We will include requirement.txt to specify the versions of all the dependencies in the code and re-upload it.
> > >
> > >
> > >
> > > **8. The proposed method without compositionality**
> > >
> > > We are still running an ablation study about the effectiveness of compositionality using the environments listed in Fig. 8a. We will post the experiment results of our method without policy ensembles, which will also be included in the updated paper.
> > >
> > >
> > >
> > > We thank the reviewer for the valuable comments and suggestions. We have been revising the paper to incorporate all of them. We will submit the updated paper shortly.

---

> > > > ### Comment · Reviewer_R2Gf · 2021-11-20
> > > > **Re: Response to Reviewer R2Gf (3/3)**
> > > >
> > > > I appreciate the authors' detailed response addressing all of my main concerns. I have updated my rating to 8 to reflect it. I am glad to see that the revised paper incorporates the reviews and I thank the authors for their thoroughness and scientific rigor demonstrated in the response.

---

> ### Author Response · Authors · 2021-11-19
> **Paper has been revised.**
>
> We would like to express our deepest gratitude for your constructive feedback again! We have revised our manuscript in response to your questions and concerns (all changed parts are colored in blue for visibility):
>
> * We have added a comparison with oracle-guided programmatic RL baselines in Sec 5 (Table 2 and the second paragraph of page 9) and Appendix C. The result confirms that our oracle-free programmatic RL overcomes the suboptimality induced by oracle-guided programmatic RL.
> * We have added a discussion on how our loop-free programmatic policies support repetitive behaviors when necessary in Appendix D. Our experiment result (Table 4 on page 19) confirms that our loop-free programs can capture repetitive behaviors and generalize better than neural policies on environment test distributions.
> * We have added an explanation as to why the convergence curves in Figure 7b are ladder-shaped at the bottom of page 8.
> * We have added a discussion about the missing related work in the first paragraph of Sec 1 and Sec 6 (page 9). We have also included a discussion on neural program synthesis in Appendix E (referred to in Sec 6).
> * We have performed an ablation study to investigate the impact of compositionality on our algorithm. The following table shows the result of comparing different strategies, i.e., learning a programmatic ensemble policy and learning a programmatic affine policy. A programmatic affine policy is a monolithic program in our DSL that switches back and forth between a set of *affine* controllers under different environmental conditions.
>
> | Environments | Programmatic Affines | Programmatic Ensembles |
> | ----------- | ----------- | ----------- |
> | Ant Cross Maze     | 0.87$\pm$0.03 | 0.10$\pm$0.06 |
> | Ant Random Goal    | 0.95$\pm$0.06 | 0.14$\pm$0.04 |
> | HalfCheetah Hurdle | 0.57$\pm$0.13 | 0.03$\pm$0.01 |
> | Pusher             | 0.36$\pm$0.05 | 0.09$\pm$0.05 |
>
> The result shows that our programmatic ensemble policies outperform the programmatic affine policies significantly in all group one environments, which highlights the merits of composition. We have included this result in Sec 5 (Table 2 and the first paragraph of page 9). Particularly, we have included the convergence curves of this ablation study in Fig. 8 of the appendix.
>
> In conclusion, we have performed new experiments and revised our manuscript following your valuable suggestions, which we feel significantly strengthen the paper. For this, we greatly appreciate your insightful comments. Please let us know if the revised paper has adequately clarified your concerns.

---

### Author Response · Authors · 2021-11-22
**Many thanks to the reviewers!**

We are grateful for the reviewers' constructive suggestions that have helped us greatly improve the quality of our manuscript. The added experiments comparing our approach against oracle-guided programmatic RL methods (suggested by all the reviewers) significantly strengthened our claim about the advantages of oracle-free programmatic RL. Particularly, we thank Reviewer R2Gf for the suggestion of rigorously investigating the impact of compositionality. It helped us demonstrate the necessity of synthesizing composite programs to solve challenging RL environments. We thank Reviewer iSsz for the suggestion of precisely analyzing the complexity of our method. It helped us make the manuscript more informative and scientifically complete. We thank Reviewer xiEe for the suggestion of thoroughly studying the sensitivity of our approach to the tree depth of our learning representation. It helped us prove the robustness of our algorithm. We also appreciate the reviewers' insightful comments that shed light on the future research directions for us!

---

### Decision · Program_Chairs · 2022-01-20

**Decision:**

Accept (Spotlight)

**Comment:**

This paper presents an approach to synthesize programmatic policies, utilizing a continuous relaxation of program semantics and a parameterization of the full program derivation tree, to make it possible to learn both the program parameters and program structures jointly using policy gradient without the need to imitate an oracle.  The parameterization of the full program derivation tree that can represent all programs up to a certain depth is interesting and novel.  In its current form this won’t scale to large programs that require large tree depth, but is a promising first step in this direction.  The learned programmatic policies are more structured and interpretable, and also demonstrated competitive performance against other commonly used RL algorithms.  During the reviewing process the authors have actively engaged in the interaction with the reviewers and addressed all the concerns, and all reviewers unanimously recommend acceptance.